

# 1 Quantify and reduce flood forecast uncertainty by the CHUP-
# 2 BMA method

Zhen Cui[1], Shenglian Guo[1*], Hua Chen[1], Dedi Liu[1], Yanlai Zhou[1], Chong-Yu Xu[2]
[1] State Key Laboratory of Water Resources and Hydropower Engineering Science, Wuhan University, Wuhan, China
[2] Department of Geoscience, University of Oslo, Oslo, Norway
*Correspondence to*: Shenglian Guo (slguo@whu.edu.cn)
**Abstract.** The Bayesian model averaging (BMA), hydrological uncertainty processor (HUP), and HUP-BMA methods have
been widely used to quantify flood forecast uncertainty. This study, for the first time, introduced a copula-based HUP in the
framework of BMA and proposed the CHUP-BMA method to bypass the need for normal quantile transformation of the
HUP-BMA method. The proposed ensemble forecast scheme consists of 8 members (two forecast precipitation inputs, two
advanced long short-term memory (LSTM) models, and two objective functions used to calibrate parameters) and is applied
to the interval basin between Xiangjiaba and Three Gorges Reservoir (TGR) dam-site. The ensemble forecast performance of
the HUP-BMA and CHUP-BMA methods is explored in the 6-168h forecast horizons. The TGR inflow forecasting results
show that the two methods can improve the forecast accuracy over the selected member with the best forecast accuracy, and
the CHUP-BMA performs much better than the HUP-BMA. Compared with the HUP-BMA method, the forecast interval
width with the 90% confidence level and continuous ranked probability score metrics of the CHUP-BMA method are highest
reduced by 28.42% and 17.86%, respectively. The probability forecast of the CHUP-BMA method has better reliability and
sharpness and is more suitable for flood ensemble forecasts, providing reliable risk information for flood control decision-
making.

## 20 1 Introduction

Accurate and reliable flood forecasting is one of the necessary measures to reduce flood disasters and improve water
resource utilization (Zhou et al., 2019; Vegad and Mishra, 2022). With the development of hydrological theory and flood
forecasting techniques, the flood forecasting accuracy and lead time have been significantly improved in recent years (Xu et
al., 2022; Cui et al., 2023). However, neither physically-based and conceptual hydrological models nor data-driven models
can guarantee to obtain perfect forecasting in real conditions. Because of the influence of the changing environment and the
limitations of human perception of complex hydrological processes, the meteorological forcing and other inputs,
hydrological model structure, and parameters, etc., contain significant uncertainties (Cloke et al., 2009), which leads to the
simulation and forecast results of the model inevitably containing integrated uncertainties from multiple sources (Liu et al.,
2022). Traditional flood forecasting schemes are mostly deterministic forecast results without considering forecast





uncertainty (Zhong et al., 2018a; Gelfan et al., 2018), which makes decision-makers unable to grasp useful risk information
beyond the forecast value. Excessive superstition on a single forecast value will likely lead to poor decision-making
(Krzysztofowicz et al., 1999). Therefore, it is essential to quantify and reduce flood forecast uncertainty in practical
applications.
Probabilistic flood forecasting is one of the effective methods to quantify integrated forecast uncertainty (Matthews et
al., 2022). It not only provides a deterministic forecast value, but also provides forecast uncertainty (or risk) information by
means of quantile, confidence interval, or density function (Biondi and Todini, 2018; Ferretti et al., 2020; Zhou et al., 2022),
which is more scientifically reasonable and practically useful compared with deterministic forecasts and helps decision-
makers consider forecast risk quantitatively (Todini, 2008). Various probabilistic forecasting methods based on statistical
post-processing of numerical forecast data have been developed in recent years. Among these methods, probabilistic
ensemble forecasting is considered to overcome the limitations of a single model or a simple average with fixed model
weights (Han and Coulibaly, 2017) and contains richer forecast information because it can consider the ensemble forecast
results of multiple models to quantify and reduce integrated uncertainty that contains uncertainties in the inputs, model
structure, and parameters (Li et al., 2017; Saleh et al., 2016). Bayesian model averaging (BMA), proposed by Raftery et al.
(2005), uses the Bayesian theory and a total probability formulation to transform ensemble forecasts into probabilistic
forecasts and is one of the most representative and reliable methods that has been widely used to supplement uncertainty
information beyond point estimates (Shu et al., 2022).
The BMA method is initially successfully applied to the ensemble forecast of meteorological elements such as
temperature and precipitation (Raftery et al. 2005; Sloughter et al., 2010). After confirming that the BMA method can
effectively quantify forecast uncertainty and obtain highly accurate deterministic forecasts, it is widely used in hydrological
forecasting to quantify forecast uncertainty from different sources, such as model inputs, structure, and parameters. The
standard BMA method assumes that each member's posterior probability distribution approximately obeys a normal
distribution (Huang et al., 2019; Guo et al., 2021). However, some variables, such as wind speed, rainfall, runoff, etc.,
usually obey skewed distributions and require methods such as Box-Cox to convert non-Gaussian variables to standard
normal variables that affect the accuracy of probability distribution estimation (Duan et al., 2007; Liu et al., 2018). Many
authors have investigated the applicability of BMA in flood ensemble forecasting and tried to overcome its limitations
(Madadgar and Moradkhani, 2014; Darbandsari and Coulibaly,2020). Sloughter et al. (2010) proposed an improved BMA
method by assuming that the posterior probability distribution of each member could obey a specific non-normal distribution
(e.g., Gamma distribution) and using the member forecast values to estimate the mean and variance of the distribution.
Madadgar and Moradkhani (2014) introduced the Copula function to solve the posterior probability distribution of members
in the BMA method and proposed the Copula-based BMA method, which avoids the assumption of the posterior probability
distribution and further reduces the application limitation of the BMA method. Meanwhile, the BMA method usually
ensembles the forecast results of multiple models to be as close to the actual values as possible. However, too many
ensemble members may generate redundant information. Darbandsari and Coulibaly (2020) introduced the Shannon entropy





theory to select the forecast members that satisfy the above conditions before applying BMA. Their results showed that the BMA method incorporating entropy could improve the probabilistic forecasting performance for high flows over the standard BMA method. In addition, some studies have developed various methods based on the BMA principle, such as the multi-model ensemble forecasting method based on Vine Copula (Zhang et al., 2022) and the combination of BMA and data assimilation techniques (Parrish et al., 2012). However, most studies ignore an essential issue: the BMA does not consider the constraint of initial conditions (i.e., observed flow at the start of the forecast). When the member forecasts are the same, the ensemble forecast will produce the same conditional probability distribution and lack rationality.

The hydrological uncertainty processor (HUP) can obtain the posterior distribution function of the actual value under the condition of the forecast value and the observed flow at the start of the forecast based on Bayesian principles and the assumption of perfect rainfall forecasting (Krzysztofowicz and Kelly, 2000). Darbandsari and Coulibaly (2021) utilized the HUP method to derive the posterior distribution of each member and used the BMA method to weight the posterior distribution of all members to obtain the final posterior distribution, which is called the HUP-BMA method. Their results showed that the HUP-BMA method outperforms the HUP method and improves the BMA method in short-term probabilistic forecasting. In addition, the derivability of the posterior distribution for the ensemble members is theoretically enhanced, the heteroskedasticity of the ensemble members is considered, and the interpretability and logical rationality of the BMA method are improved

Although it has been demonstrated that considering initial conditions in the BMA method can improve ensemble forecast performance, there are still issues to be explored. The HUP-BMA method requires a normal quantile conversion method to convert the flow data series to Gaussian space to solve the posterior distribution. The process is not only tedious and complicated, but also prone to bias in the inverse conversion. To this end, Liu et al. (2018) proposed a Copula-based HUP (CHUP) and found that it could bypass the normal transformation process and improve the probabilistic forecasting. It is anticipated that coupling CHUP to the BMA may improve the HUP-BMA accuracy and applicability, which motivates the current study.

Therefore, the main aims and research steps of this study are as follows: (1) to propose the novel CHUP-BMA method by coupling CHUP into BMA to realize the consideration of the initial condition of the forecast while bypassing the need for data normal quantile transformation; (2) to construct an ensemble forecast containing 8 members combining two types of forecast precipitation, two long short-term memory (LSTM) models, i.e., the recursive encoder-decoder structure-based LSTM-RED model and the feature-temporal dual attention-based DA-LSTM-RED model, and two objective functions of model calibration; and (3) to analyze and discuss the ensemble forecast performance of the proposed method in terms of the deterministic and probabilistic forecast as compared with the HUP-BMA benchmark method. The interval basin between Xiangjiaba Dam and the Three Gorges Dam is selected for the proposed study.

The rest of the paper is organized as follows. Section 2 introduces the case study and materials. The methods are presented in Section 3. Section 4 evaluates the deterministic and ensemble forecast results. Conclusions and prospects are given in Section 5.





## 2 Case study and materials

### 2.1 Study basin

Three Gorges Reservoir (TGR) is the largest hydraulic project in the world and plays a vital role in flood control, power generation, and other water resource management issues (Zhong et al., 2020). The TGR controls a watershed area of about 1 million km$^2$. The total reservoir capacity is about 39.3 billion m$^3$, with a flood control capacity of about 22.15 billion m$^3$.

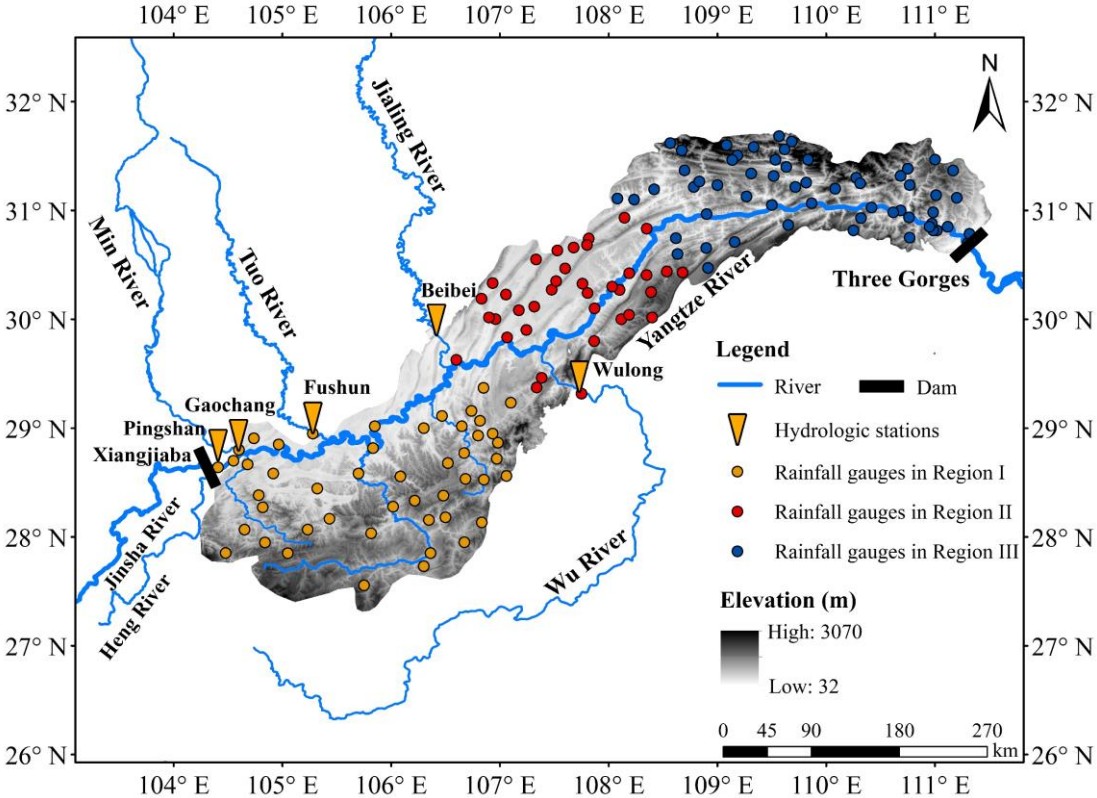

**Fig. 1** Schematic diagram of the interval-basin between Xiangjiaba and TGR dam-site which is divided into three sub-regions.

The TGR inflow is directly influenced by the runoff yield of the cascade reservoir interval-basin between Xiangjiaba and TGR (Fig.1), with a basin area of about 127,400 km$^2$ (Zhou et al., 2019). The inflow of the TGR consists of the outflow discharge from the Xiangjiaba Reservoir, the inflow of several tributaries such as Min, Tuo, Jialing, and Wu Rivers, and the rainfall of the interval-basin. The flow sources are complex and have different effects on the TGR inflow. Moreover, TGR is a river-type reservoir with a length of about 600 km at the normal storage level (175m) and an average width of only 1.1 km, resulting in uncertainty in rainfall intensity and storm-centre positioning (Zhong et al., 2020). Therefore, there is significant uncertainty in the flood forecast of TGR. It has been a major challenge to quantify and reduce forecast uncertainty.





Table 1 shows the flow propagation time from the hydrological control stations of the mainstream and tributaries to the
TGR dam. The outflow discharge of Xiangjiaba Reservoir, located on the Jinsha River, is observed at the Pingshan
hydrological station and represents the mainstream flow. The discharge values from large tributaries (Min, Jialing, Tuo, and
Wu Rivers) are observed at the Gaochang, Fushun, Beibei, and Wulong hydrological stations, respectively.
Considering the uneven distribution of rainfall intensity because of the narrow and long basin, the interval basin
between Xiangjiaba and TGR dam-site is divided into three sub-basins: Pingshan-Cuntan, Cuntan-Wanxian, and Wanxian-
TGR dam-site. Their watershed areas are 76,900, 22,900, and 27,600 km2 respectively. Meanwhile, there are 45, 38, and 60
gauged rainfall stations in these three sub-regions, respectively.

**Table 1** List of flow propagation time for hydrological control stations to TGR dam-site

| Rivers | Hydrological control stations | To TGR dam flow propagation time (h) |
|---|---|---|
| Jinsha | Pingshan | 48-66 |
| Min | Gaochang | 48-66 |
| Jialing | Beibei | 24-42 |
| Tuo | Fushun | 42-60 |
| Wu | Wulong | 15-30 |


## 2.2 Study materials

This study collects 6h observed flow discharges at TGR dam-site and five hydrological stations (Table 1), and 6h
observed rainfall in the interval-basin during the 2010-2021 flood season (May-September). The Tyson polygon method is
used to calculate areal average rainfall using rainfall station data for each sub-basin area. Meanwhile, this study collects the
forecasted precipitation data issued by the European Centre for Medium-Range Weather Forecasts (ECMWF) and the
Hydrology Bureau of the Yangtze River Water Resources Commission (HBYRWRC) for the 2017-2021 flood season in the
three sub-basins. Their forecast time starts at 8:00, with the 6-168h forecast horizons and the 6h forecast interval. The spatial
resolution of each grid for the ECMWF forecasted precipitation is 0.125°×0.125°. The HBYRWRC forecasted precipitation
is the areal average forecasted precipitation data.
The training period is from 2010 to 2016, and the validation period is from 2017 to 2021. Since the precipitation
forecast starts at 8:00 a.m., the forecasted flow for the 6-168h forecast horizons is also calculated from the daily 8:00 a.m. in
the validation period.



## 3 Methods

### 3.1 Proposed CHUP-BMA method

#### 3.1.1 Bayesian model averaging (BMA)

Bayesian model averaging (BMA) method's principle is as follows.

$$p(Q_o|Q_{f,1}, Q_{f,2}, \dots, Q_{f,k}) = \sum_{i=1}^{k} w_i \cdot p(Q_o|Q_{f,i}) \tag{1}$$

where, where, $p(\cdot)$ denotes the probability density function. $Q_o$ denotes the observed flow corresponding to the forecast moment (target value). $k$ is the number of ensemble members. $Q_f$ denotes the forecasted flow of ensemble members. $w_i$ denotes the weight of the $i$-th model. $p(Q_o/Q_{f,i})$ denotes the conditional probability density of $Q_o$ conditional on $Q_{f,i}$, which is assumed to approximately obey a normal distribution with the expectation of $\mu_{i} = a_i + b_i \cdot Q_{f,i}$ and variance of $\sigma_i$. $a_i$ and $b_i$ are the bias correction coefficients obtained by linear fitting of $Q_{f,i}$ to $Q_o$.

Therefore, Eq. (1) can be rewritten as follows.

$$p(Q_o|Q_{f,1}, Q_{f,2}, \dots, Q_{f,k}) = \sum_{i=1}^{k} w_i \cdot N(Q_o|\mu_i, \sigma_i) \tag{2}$$

From Eq. (2), it can be seen that the BMA method does not consider the influence of the initial state (the actual observed flow at the start of the forecast) on the posterior distribution. When the member forecasts at different times are the same, the posterior probability distribution generated by the BMA is also the same, which lacks logical rationality.

#### 3.1.2 Hydrological uncertainty processor (HUP)

Based on the assumption that the precipitation uncertainty is zero, under the condition that the $i$-th ensemble member forecasts ($Q_{f,i}$) and the observed flow at the start of the forecast ($Q_b$), the posterior distribution of $Q_o$ derived by the HUP method is as follows.

$$p(Q_o|Q_{f,i}, Q_b) = \frac{p(Q_{f,i}|Q_o, Q_b) \cdot p(Q_o|Q_b)}{\int_{-\infty}^{+\infty} p(Q_{f,i}|Q_o, Q_b) \cdot p(Q_o|Q_b) dQ_o} \tag{3}$$

where, $p(Q_o/Q_b)$ is the prior density function, $p(Q_{f,i}/Q_o, Q_b)$ is the likelihood density function. $p(Q_o/Q_{f,i}, Q_b)$ is the posterior density function.

The HUP method is a meta-Gaussian model assuming that the runoff series obeys a normal distribution, the core of which is the normal quantile transformation (Liu et al., 2016).

$$\hat{Q}_o = N^{-1}(P(Q_o)), \hat{Q}_{f,i} = N^{-1}(P(Q_{f,i})) \tag{4}$$

where, $P(\cdot)$ denotes the probability distribution function. $N^{-1}(\cdot)$ denotes the inverse function of the standard normal distribution. $\hat{Q}_o$ and $\hat{Q}_{f,i}$ are the observed and forecasted flow transformed to the normal space, respectively.





The HUP method assumes that the observed flow obeys a first-order Markov process (Krzysztofowicz and Kelly, 2000),
i.e., the flows between adjacent forecast horizons obey the linear constraint after the normal transformation.

$$\hat{Q}_{o,t} = c_t \times \hat{Q}_{o,t-1} + \varepsilon_t \tag{5}$$

where, $\hat{Q}_{o,t}$ is the observed flow corresponding to the $t$-th forecast horizon. $c$ is the regression coefficient. $\varepsilon$ is the residual,
obeying $N(0,1-c_t^2)$.
The prior density function expressions are as follows.

$$p(\hat{Q}_{o,t}|\hat{Q}_b) = \frac{1}{(1-C_t^2)^{0.5}} n\left\{\frac{\hat{Q}_{o,t} - C_t \times \hat{Q}_b}{(1-C_t^2)^{0.5}}\right\}, C_t = \prod_{i=1}^{t} c_i \tag{6}$$

where, $n(\cdot)$ denotes standard normal density function; $\hat{Q}_b$ is the observed flow at the start of the forecast transformed to the
normal space.
$\hat{Q}_b$, $\hat{Q}_o$, and $\hat{Q}_{f,i}$ are assumed to obey a linear relationship. The expression of the likelihood function in normal space is
as follows.

$$p(\hat{Q}_{f,i,t}|\hat{Q}_{o,t}, \hat{Q}_b) = \frac{1}{\sigma_t} n\left\{\frac{\hat{Q}_{f,i,t} - (a_t \times \hat{Q}_{o,t} + d_t \times \hat{Q}_b + b_t)}{\sigma_t}\right\} \tag{7}$$

where, $\theta_t$ is an independent variable obeying $N(0,\sigma_t^2)$. $a_t$, $d_t$, and $b_t$ are regression coefficients.
The posterior density function under normal space can be derived by substituting Eqs. (6) and (7) into Eq. (3).

$$p(\hat{Q}_{o,t}|\hat{Q}_{f,i,t}, \hat{Q}_b) = \frac{1}{Y_t} n\left\{\frac{\hat{Q}_{o,t} - (A_t \times \hat{Q}_{f,i,t} + D_t \times \hat{Q}_b + B_t)}{Y_t}\right\},$$

$$A_t = \frac{a_t y_t^2}{a_t^2 y_t^2 + \sigma_t^2}, B_t = \frac{-a_t b_t y_t^2}{a_t^2 y_t^2 + \sigma_t^2}, D_t = \frac{C_t \sigma_t^2 - a_t d_t y_t^2}{a_t^2 y_t^2 + \sigma_t^2}, Y_t = \left(\frac{y_t^2 \sigma_t^2}{a_t^2 y_t^2 + \sigma_t^2}\right)^{0.5}, y_t^2 = 1 - C_t^2 \tag{8}$$

The posterior distribution function under the normal space can be converted to the original space by Jacobian
transformation (Liu et al., 2016). The posterior density function of $Q_{o,t}$ under $Q_{f,i,t}$ and $Q_b$ conditions is as follows.

$$p(Q_{o,t}|Q_{f,i,t}, Q_b) = \frac{J(Q_{o,t})}{Y_t} n\left\{\frac{N^{-1}(P(Q_{o,t})) - A_t N^{-1}(P(Q_{f,i,t})) - D_t N^{-1}(P(Q_b)) - B_t}{Y_t}\right\},$$

$$J(Q_{o,t}) = \frac{p(Q_{o,t})}{n\left(N^{-1}(P(Q_{o,t}))\right)} \tag{9}$$

where, $J(\cdot)$ is the Jacobian transformation function.

### 3.1.3 HUP-BMA method

Darbandsari and Coulibaly et al. (2021) applied the hydrological uncertainty processor (HUP) to the ensemble forecast
members, substituted the posterior density function obtained by the HUP method (Eq. (9)) into the BMA framework (Eq.





(2)), and then obtained the posterior distribution function of the target flow based on the initial state and the forecasted flow
of the ensemble member. Therefore, the expression of the HUP-BMA method is as follows.

$$p(Q_o|Q_{f,1}, Q_{f,2}, \ldots, Q_{f,k}, Q_b) = \sum_{i=1}^{k} w_i \cdot \frac{J(Q_{o,t})}{Y_t} n \left\{ \frac{N^{-1}\left(P(Q_{o,t})\right) - A_t N^{-1}\left(P(Q_{f,i,t})\right) - D_t N^{-1}(P(Q_b)) - B_t}{Y_t} \right\} \quad (10)$$

### 179  3.1.4 Copula-based HUP-BMA (CHUP-BMA) method

#### 180  (1) Copula-based HUP

According to Sklar's theorem (Sklar, 1959), the joint distribution of $m$ variables is as follows.

$$P(x_1, x_2, \ldots \ldots, x_m) = C_m(P(x_1), P(x_2), \ldots \ldots, P(x_m)) \quad (11)$$

where, $C_m(\cdot)$ denotes the m-dimensional copula distribution.
The copula-based HUP method (CHUP) was proposed by Liu et al. (2018), which can avoid the normal quantile
transformation process of the flow series in the standard HUP method. With the help of the copula function, the prior density
function in Eq. (3) can be derived as follows.

$$p(Q_o|Q_b) = \frac{\partial^2 C_2(P(Q_o), P(Q_b))}{\partial P(Q_o)\partial P(Q_b)} \cdot \frac{dP(Q_o)}{dQ_o} = c_2(P(Q_o), P(Q_b)) \cdot p(Q_o) \quad (12)$$

where, $c_m(\cdot)$ denotes the m-dimensional copula density function.
The likelihood density function in Eq. (3) can be derived as follows.

$$p(Q_{f,i}|Q_o, Q_b) = \frac{\dfrac{\partial^3 C_3(P(Q_o), P(Q_{f,i}), P(Q_b))}{\partial P(Q_o) \cdot \partial P(Q_{f,i}) \cdot \partial P(Q_b)}}{\dfrac{\partial^2 C_2(P(Q_o), P(Q_b))}{\partial P(Q_o) \cdot \partial P(Q_b)}} \cdot \frac{dP(Q_{f,i})}{dQ_{f,i}} = \frac{c_3(P(Q_o), P(Q_{f,i}), P(Q_b))}{c_2(P(Q_o), P(Q_b))} \cdot p(Q_{f,i}) \quad (13)$$

The posterior density function in Eq. (3) can be derived as follows.

$$p(Q_o|Q_{f,i}, Q_b) = \frac{c_3(P(Q_o), P(Q_{f,i}), P(Q_b))}{\int_0^1 c_3(P(Q_o), P(Q_{f,i}), P(Q_b))dP(Q_o)} \cdot p(Q_o) \quad (14)$$

#### 189  (2) Copula-based HUP-BMA method

Applying CHUP to the $i$-th ensemble member, the posterior probability distribution function $p(Q_o|Q_{f,i}, Q_b)$ of $Q_o$ based
on $Q_{f,i}$ and $Q_b$ can be obtained. Coupling $p(Q_o|Q_{f,i}, Q_b)$ into the BMA framework, the copula-based HUP-BMA (CHUP-
BMA) method can be constructed, and Eq. (2) can become as follows.

$$p(Q_o|Q_{f,1}, Q_{f,2}, \ldots, Q_{f,k}, Q_b) = \sum_{i=1}^{k} w_i \cdot \frac{c_3(P(Q_o), P(Q_{f,i}), P(Q_b))}{\int_0^1 c_3(P(Q_o), P(Q_{f,i}), P(Q_b))dP(Q_o)} \cdot p(Q_o) \quad (15)$$

The forecast uncertainty is quantified by the forecast interval with a 90% confidence level. Before constructing the
copula, selecting the marginal distribution and the copula type is usually necessary. This study intends to select the
appropriate marginal distribution and copula function from five common distribution functions, such as Pearson type III (P-



III), Gamma, Normal, Lognormal, and Weibull, and five common copula functions, such as Gumbel-Hougaard, Frank,
Clayton, Student-t (Student) and Gaussian copula, according to the root mean square error (RMSE) minimization criterion,
respectively. The definition and mathematical expressions of copula functions can be referred to Liu et al. (2018) and Chen
and Guo (2019).
Darbandsari and Coulibaly (2021) demonstrated that the HUP-BMA method could improve the probabilistic forecasting
performance of the HUP and BMA methods in the short forecast horizons. Therefore, this paper focuses on analyzing and
comparing the performance of the HUP-BMA and CHUP-BMA methods. The HUP-BMA and CHUP-BMA methods only
calibrate the ensemble members' weights through the Expectation-Maximization (EM) algorithm (Darbandsari and
Coulibaly, 2021). Meanwhile, since the forecast accuracy of ensemble members may change with time due to seasonality
and other factors (Zhong et al., 2020), the sliding window approach is used to update the weighting parameters. It has been
studied that the BMA method with sliding windows can obtain better probabilistic forecast performance (Parrish et al., 2012;
Darbandsari and Coulibaly, 2019).
**3.2 Ensemble forecasting scheme**
An ensemble forecast scheme containing multi-source uncertainties in the model input, the model structure, and the
parameter is constructed using a multi-member approach consisting of two forecasted precipitation, two models, and two
objective functions used to calibrate parameters, as shown in Fig. 2.

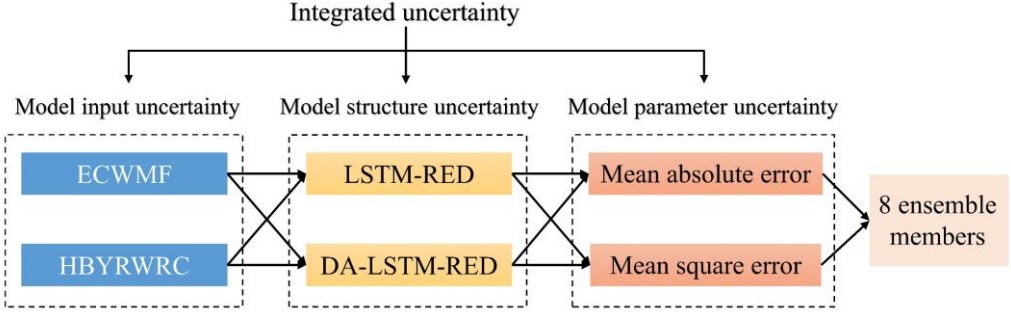

**Fig. 2** The TGR's flood ensemble forecast scheme

**3.2.1 Model input uncertainty**
This study adopts forecasted precipitation released by ECMWF and HBYRWRC to consider the uncertainty of model
inputs. The TGR is a river-type reservoir, so building a river confluence model for flood forecasting is necessary. The
observed and forecasted precipitations are converted to effective precipitation, i.e., the precipitation after the loss of plant
retention, infiltration, evaporation, etc., to consider the runoff yield in the three sub-basin areas. The rainfall-runoff





relationship graph method (Fedora and Beschta, 1989) commonly used in the Yangtze River basin (Fig. 3) calculates the
effective precipitation in the three sub-basin areas. The core of this method is to calculate the antecedent precipitation index
representing soil moisture (Zhong et al., 2018b), shown in the following equation.

$$P_{a,t+1} = k(P_{a,t} + P_t) \tag{16}$$

$$P_{a,t+1} \leq I_m \tag{17}$$

where, $P_a$ denotes the antecedent precipitation index. $P_t$ is the daily precipitation. $I_m$ is the water storage capacity of the basin.
$k$ denotes evaporation reduction index. The $k$ and $I_m$ for the three sub-basin areas are shown in Table 2.

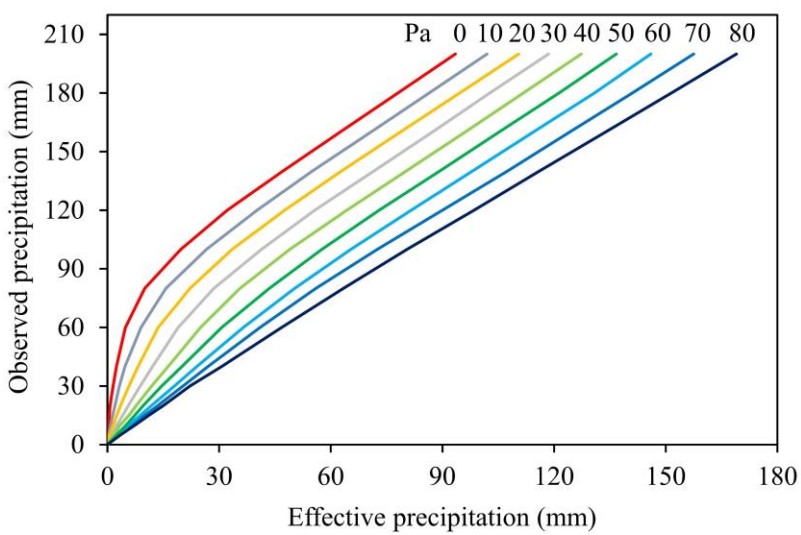


**Fig. 3** Rainfall-runoff relationship between Xiangjiaba and Three Gorges Dam-site uncontrolled interval basin

**Table 2** The $k$ and $I_m$ values for the three sub-basin areas

| Sub-basin | $k$ | $Im$ |
|---|---|---|
| Pingshan-Cuntan | 0.90 | 50 |
| Cuntan-Wanxian | 0.95 | 80 |
| Wanxian-TGR dam-site | 0.95 | 80 |


After obtaining the daily antecedent precipitation index at 8:00, the antecedent precipitation index for the 6-h time scale
is calculated as follows.

$$P_{a,t,m} = \left(P_{a,t} + \sum P_{t,n}\right) \times k^{\frac{h}{24}} \tag{18}$$

where, $P_{a,t,m}$ denotes the antecedent precipitation index at $m$:00 on the $t$-th day. $\sum P_{t,n}$ denotes the cumulative observed
precipitation from 8:00 to $m$:00 on the $t$-th day. $h$ denotes the time gap from 8:00 to $m$:00 on the $t$-th day.



### 3.2.2 Model structure uncertainty

The TGR inflow forecasting is influenced by the upstream mainstream and tributary reservoir scheduling decisions, the rainfall intensity and distribution in the interval basin, and the changes in the subsurface characteristics, which is challenging to establish complex and physical-based hydrological models (Yang et al., 2019; Cho et al., 2022; Hauswirth et al., 2023). The simulation or forecast accuracy in this interval-basin needs to be improved to meet the needs of the work. Therefore, two advanced data-driven models for obtaining multi-step-ahead flood processes forecasting, namely the long short-term memory (LSTM-RED) model based on an encoder-decoder structure and the coupled dual attention LSTM-RED (DA-LSTM-RED) model, are used for confluence calculations as a way to consider the uncertainty in the model structure.

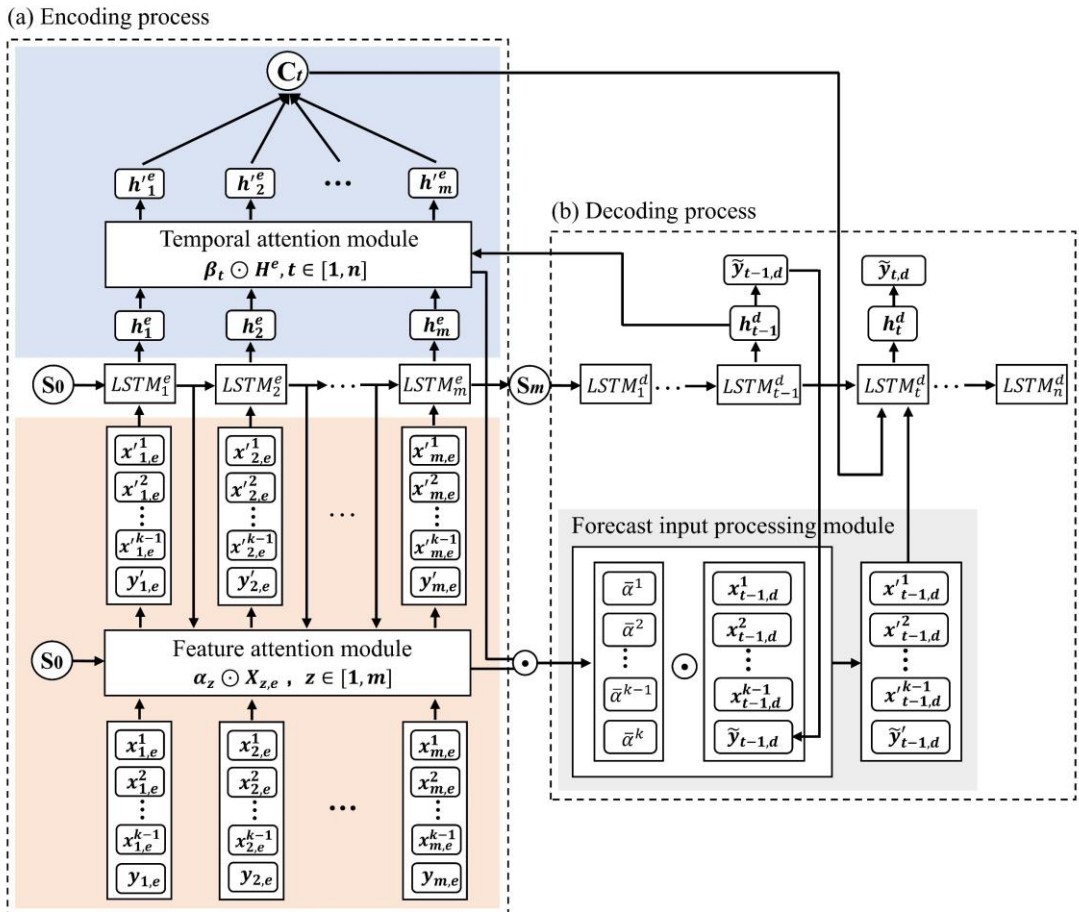

**Fig. 4** Schematic diagram of the DA-LSTM-RED model. $e$ and $d$ are the encoding and decoding processes, respectively. $k$ is the number of input types. $\mathbf{X_{z,e}}$ is the input variables of encoding process, $\mathbf{X_{z,e}}=\{x_{z,e}^1, x_{z,e}^2, ..., x_{z,e}^{k-1}, y_{z,e}\}$. $\boldsymbol{\alpha_z}$ denotes the weights of the input variables, $\boldsymbol{\alpha_z}=\{\alpha_z^1, \alpha_z^2, ..., \alpha_z^k\}$. $m$ is the input time-steps in the encoding process. $S$ is the hidden layer output. $n$ is the maximum forecast horizon. $\boldsymbol{H^e}$





is the hidden layer state, $\boldsymbol{H}^e=\{h_1^e, h_2^e, ..., h_m^e\}$. $\boldsymbol{\beta}_t$ denotes the weights of the hidden layer states of the encoding process, $\boldsymbol{\beta}_t=\{\beta_t^1, \beta_t^2, ..., \beta_t^m\}$.
$C$ denotes the key information highlighted by the temporal attention. $\bar{\alpha}$ denotes the forecast input weights.

**(1) Long short-term memory model based on encoder-decoder structure**
This study nests an LSTM neural network into a recursive encoder-decoder (RED) structure that can be obtained for
forecasting flood processes to build an LSTM-RED model. Among them, the RED structure is similar to that of Kao et al.
(2020). The description of the LSTM neural network can be found in Cui et al. (2022). The encoding process of the RED
structure is used to extract the critical information ($C_t$) of the input (Xiang et al., 2020). In the decoding process, forecast
information of the same category as the encoding process is input to neural network of the latter moment, besides the $C_t$ and
the output of the hidden layer at the previous moment.
**(2) LSTM-RED neural network coupled dual attention mechanism**
The LSTM-RED model based on the dual attention mechanism (DA-LSTM-RED) is established by adding the feature-
temporal dual attention mechanism to the LSTM-RED model, which can enable the model to highlight effective information
in different types and moments of the input. The DA mechanism (Fig. 4) consists of the feature attention module, the
temporal attention module, and the forecast input processing module.
The feature attention module can adaptively highlight the critical input types by assigning feature weights to the input
of the encoding process (Qin et al., 2017). The temporal attention module can highlight the information (hidden layer states)
extracted at critical time step by assigning temporal weights to the information extracted at all time step in the encoding
process (Ding et al., 2020). Meanwhile, the feature weights are averaged based on temporal weights and applied them to the
forecast information inputted in the decoding process, thus highlighting the key forecast input variables. The principle of the
DA-LSTM-RED model can be found in Cui et al. (2023).
**(3) Model input and hyperparameter selection**
In this study, the input types for the encoding process include effective precipitation in the three sub-basins, flow
discharge in the mainstream and tributaries (i.e., five hydrological stations in Table 1), and previously observed inflow to the
TGR for a total of nine types data. In order to make the model learn comprehensive information, input variables with the last
11-time steps (66h) are inputted to the encoding process according to the flow propagation times from the hydrological
stations to the TGR dam site in Table 1.
The forecasted effective precipitation, the forecasted flow of the mainstream and tributaries, and the forecasted flow for
the previous forecast horizon are used as inputs of the decoding process. Among them, the forecasted effective precipitation
is calculated by the observed precipitation during the training period and by the forecast precipitation during the validation
period. The forecasted flow of the mainstream and tributaries is replaced by the observed flow during the training and
validation periods. The TGR's observed inflow for the 6-168h forecast horizons is the target output, needed for practical
forecasting.





The input and output data are handled by the normalization method. Moreover, the trial-and-error method is used for debugging the network hyperparameters. The model is trained by the Adam method.

**3.2.3 Model parameter uncertainty**

Different parameter-optimization objective functions may focus on different forecast results (Zhong et al., 2020). For example, the mean absolute error function considers flow errors of different magnitudes equally. The mean square error function usually magnifies the errors in the high flow, which may make the model parameters with different objective functions produce forecast results with different focus points (Duan et al., 2007). Therefore, it is necessary to consider the uncertainty of the model parameters. Neural network models usually train model parameters (such as model internal weights and bias values, etc.) based on loss functions, so this paper uses two common loss functions, namely the mean absolute error and the mean square error, to train the model as a way to consider the uncertainty of model parameters.

**3.3 Evaluation metrics**

**3.3.1 Deterministic forecast evaluation metrics**

The accuracy of deterministic forecast is evaluated by three metrics: the Nash-Sutcliffe efficiency (Nash and Sutcliffe, 1970) (NSE), the mean absolute error (MAE) and the relative error of total runoff (RE).

$$NSE = 1 - \frac{\sum_{i=1}^{N}(Q_{o,i} - Q_{f,i})^2}{\sum_{i=1}^{N}(Q_{o,i} - \overline{Q_o})^2} \tag{19}$$

$$RE = \frac{\sum_{i=1}^{N} Q_{f,i} - \sum_{i=1}^{N} Q_{o,i}}{\sum_{i=1}^{N} Q_{o,i}} \times 100\% \tag{20}$$

$$MAE = \frac{1}{N}\sum_{i=1}^{N}|Q_{o,i} - Q_{f,i}| \tag{21}$$

where, $N$ is the sample number. $\overline{Q_o}$ and $\overline{Q_f}$ are the average of the observed and forecasted flow, respectively.

**3.3.2 Probabilistic forecast evaluation metrics**

**(1) Forecast interval evaluation metrics**

The forecast interval is evaluated by three metrics: the average coverage cate (CR), average interval width (IW), and the percentage of observations bracketed by the unit confidence Interval (PUCI) (Li et al., 2011).

$$CR = \frac{n_c}{N} \tag{22}$$

$$IW = \frac{1}{N}\sum_{i=1}^{N}(Q_{u,i} - Q_{l,i}) \tag{23}$$





$$PUCI = \frac{CR}{\frac{1}{N}\sum_{i=1}^{N}\left(\frac{Q_{u,i} - Q_{l,i}}{Q_{o,i}}\right)} \tag{24}$$

where, $n_c$ denotes the number of $Q_o$ located in the forecast interval. $Q_u$ and $Q_l$ are the upper and lower boundaries of the
forecast interval with a 90% confidence level, respectively.
**(2) Probabilistic forecast evaluation metrics**
The probabilistic forecast is evaluated by three metrics: the α_index (Renard et al., 2010), the ignorance score (IGS)
(Gneiting et al., 2005), and continuous ranked probability score (CRPS) (Raftery et al., 2005).

$$\alpha\_index = 1 - \frac{2}{N}\sum_{i=1}^{N}|q_{e,i} - q_{th,i}| \tag{25}$$

$$IGS = -\frac{1}{N}\sum_{i=1}^{N}ln(p(Q_{o,i})) \tag{26}$$

$$CRPS = \frac{1}{N}\sum_{i=1}^{N}\int_{0}^{+\infty}\left(P_i(r) - I(r - Q_{o,i})\right)^2 dr,$$

$$I(r - Q_{o,i}) = \begin{cases} 1 & r \geq Q_{o,i} \\ 0 & r < Q_{o,i} \end{cases} \tag{27}$$

where, $q_{e,i}$ and $q_{th,i}$ denote observed and theoretical p-values of $Q_{o,i}$, respectively. p-value denotes the posterior probability
distribution value of the $Q_{o,i}$ (Renard et al., 2010). $I(\cdot)$ denotes the indicative function. $r$ denotes the flow variable.
The α_index metric can indicate the probabilistic forecast reliability and quantitatively evaluate the difference between
the quantile-quantile (Q-Q) graph curve and the 1:1 line (Thyer et al., 2009; Laio and Tamea, 2007). The closer the α_index
value is to 1, the more reliable the probabilistic forecast is. The IGS metric indicates the sharpness of the probabilistic
forecast (Gneiting et al., 2005). The lower the IGS value, the better the sharpness and the lower the forecast uncertainty. The
CRPS metric can reflect the reliability and sharpness of the probabilistic forecast and indicate the fit performance between
the posterior probabilistic distribution and the actual probabilistic distribution of $Q_o$ (Raftery et al., 2005).
**4 Results evaluation**
**4.1 Deterministic forecast results of ensemble member**
Since the study focuses on the differences in ensemble forecast performance between the HUP-BMA and CHUP-BMA
methods, the overall forecast accuracy of members is analysed (Fig. 5), and the differences in forecast accuracy between
members are not explicitly analysed. As shown in Fig. 5, using the observed values as input during the training period, high
forecast accuracy can be acquired in different forecast horizons, with the NSE values exceeding 0.95 and the MAE values
below 1400 m³/s, and the absolute value of RE within 4%.






**Fig. 5** Statistical chart of evaluation metrics of 8 ensemble members





After combining the forecasted precipitation during the validation period, the NSE values show a decreasing trend, and
the MAE and RE values show an increasing trend with the increase of the forecast horizon. Taking the NSE metrics of the 1-
7d forecast horizons as an example (Table 3), the average value of the NSE metric decreases from 0.97 to 0.89, which
indicates that the forecast accuracy gradually decreases. Meanwhile, the range of evaluation metrics gradually increases with
the increase of the forecast horizon. It can be seen from Table 3 that the difference between the maximum and minimum
values of NSE indicators for the 1d forecast horizon is only 0.01. In contrast, the difference for the 7d forecast horizon is as
high as 0.05, which indicates that the difference in forecast accuracy of members is also more significant, and the forecast
uncertainty gradually increases. Overall, the NSE values of the forecast members in the 6-168h forecast horizons are higher
than 0.88, and the absolute values of the RE metrics are within 7%. Hence, the forecast accuracy of members is relatively
high, and the forecast error is low, which can be used for flood ensemble forecasting.

336        **Table 3** Mean, minimum, and maximum values of NSE metrics for 8 ensemble members in the validation period

| Forecast horizon (h) | Mean | Max | Min | Forecast horizon (h) | Mean | Max | Min |
|---|---|---|---|---|---|---|---|
| 6 | 0.96 | 0.97 | 0.96 | 90 | 0.93 | 0.95 | 0.91 |
| 12 | 0.97 | 0.97 | 0.96 | 96 | 0.93 | 0.95 | 0.91 |
| 18 | 0.97 | 0.98 | 0.97 | 102 | 0.92 | 0.94 | 0.90 |
| 24 | 0.97 | 0.97 | 0.97 | 108 | 0.92 | 0.94 | 0.91 |
| 30 | 0.96 | 0.97 | 0.95 | 114 | 0.93 | 0.95 | 0.91 |
| 36 | 0.96 | 0.97 | 0.95 | 120 | 0.92 | 0.94 | 0.90 |
| 42 | 0.96 | 0.97 | 0.95 | 126 | 0.91 | 0.93 | 0.89 |
| 48 | 0.96 | 0.96 | 0.95 | 132 | 0.91 | 0.93 | 0.90 |
| 54 | 0.94 | 0.95 | 0.93 | 138 | 0.92 | 0.94 | 0.90 |
| 60 | 0.94 | 0.95 | 0.93 | 144 | 0.91 | 0.94 | 0.89 |
| 66 | 0.95 | 0.96 | 0.93 | 150 | 0.90 | 0.93 | 0.88 |
| 72 | 0.94 | 0.96 | 0.93 | 156 | 0.90 | 0.93 | 0.89 |
| 78 | 0.93 | 0.95 | 0.92 | 162 | 0.91 | 0.93 | 0.89 |
| 84 | 0.93 | 0.95 | 0.92 | 168 | 0.91 | 0.93 | 0.88 |


### 4.2 Ensemble forecast results

### 4.2.1 Marginal distribution and copula function selection

It is necessary first to fit the marginal distributions of the observed flow and the forecasted flow of the 6~168h forecast
horizons. The $Q_o$ and $Q_b$ obey the same distribution. The RMSE criterion is used to select the marginal distribution type. In



each forecast horizon, the RMSE values of the 8 members are averaged to obtain the marginal distribution suitable for the
forecasted flow intuitively. Meanwhile, according to Eq. (14), the three-dimensional joint distribution of $Q_o$, $Q_b$, and $Q_f$
needs to be constructed. The RMSE criterion is used to select the copula function. Similarly, the RMSE values for the 8
members of each forecast horizon were averaged.
Figs. 6 (a) and (b) show the RMSE values generated by fitting the marginal distribution and copula function,
respectively. It can be seen from Fig. 6(a) that the Lognormal distribution has the lowest RMSE value among the five
alternative marginal distributions and is chosen as the sequence marginal distribution type. As shown in Fig. 6(b), the student
copula has the lowest RMSE value in the 6-168h forecast horizons and is chosen to construct the three-dimensional joint
distribution function of $Q_o$, $Q_b$, and $Q_f$.

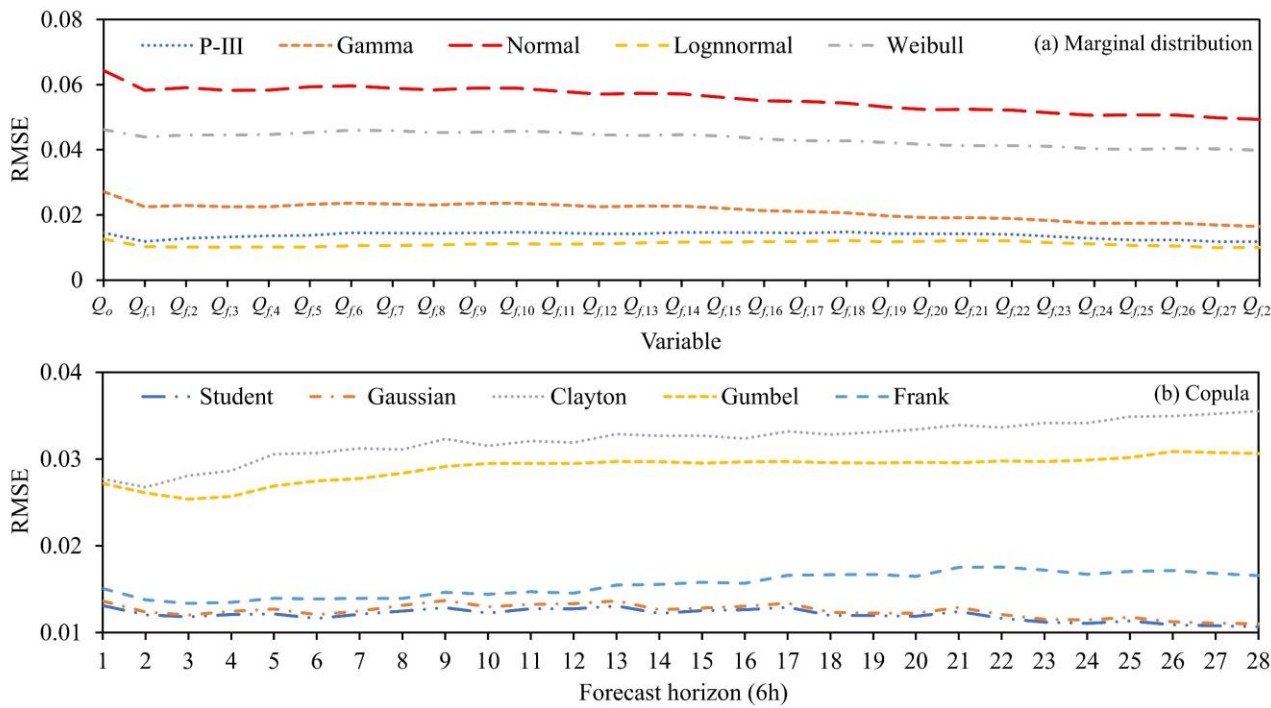


**Fig. 6** The RMSE values of $Q_o$, $Q_b$ and $Q_f$ sequence marginal distributions and copula functions. 1, 2, … ,28 denote 6h, 12h, …, 168h
forecast horizons, respectively.

### 4.2.2 Sliding window length selection

Since there is no specific method or rule to calculate the sliding window length, this study adopts the CRPS metric as
the objective function and the trial-and-error method to select the sliding window length. The range of window lengths is [40,
200]. To facilitate the selection of the sliding window lengths, Fig. 7 shows the average CRPS values of the HUP-BMA and
CHUP-BMA methods for all forecast horizons with different window lengths. It can be seen from Fig. 7 that the HUP-BMA




and CHUP-BMA methods all have the lowest CRPS values at the sliding window length of 80. Therefore, 80 is the optimal
window length for the ensemble forecasting study.

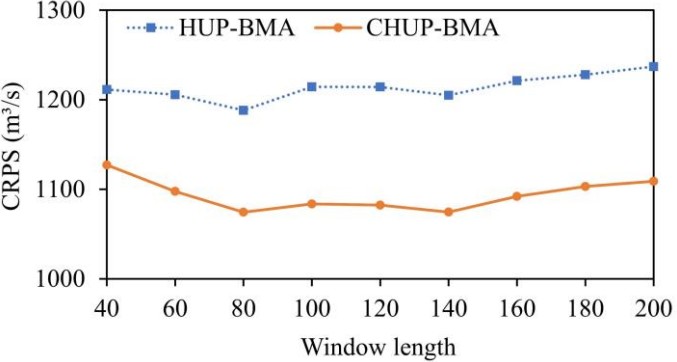


**Fig. 7** The average CRPS values of the CHUP-BMA and HUP-BMA methods with different window lengths

**4.2.3 Deterministic forecast results of ensemble forecast**
The HUP-BMA and CHUP-BMA methods use expectation values of ensemble forecasts as deterministic forecast
results. In order to analyze the deterministic forecast performance of ensemble forecasts, one member with the best forecast
accuracy is selected for comparative analysis based on the criteria of the relatively low RE and MAE values and relatively
high NSE values.

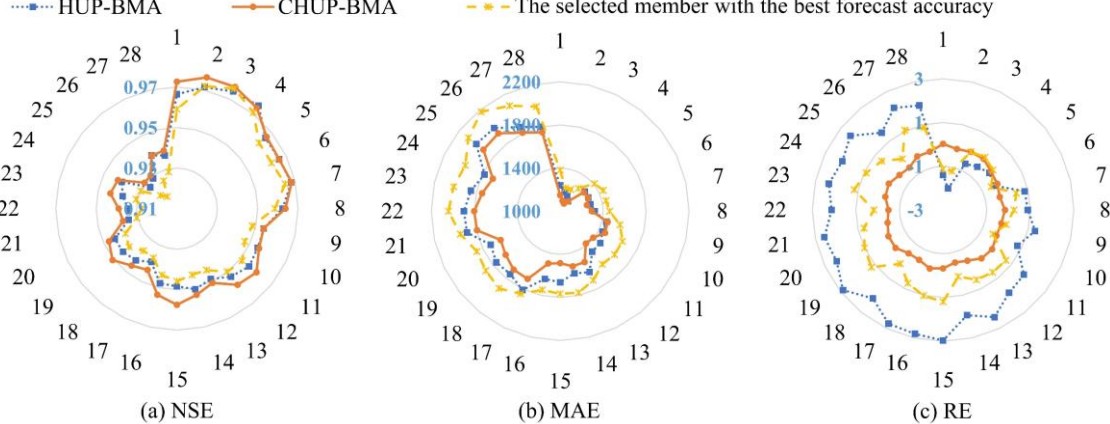


**Fig. 8** Deterministic forecast evaluation metrics for the HUP-BMA, the CHUP-BMA, and the selected member with the best forecast
accuracy





Fig. 8 (a), (b), and (c) show the NSE, MAE, and RE metrics of three deterministic forecast results, respectively. It can
be seen that the NSE metrics show a decreasing trend, and the MAE metrics show an increasing trend as the forecast horizon
increases, indicating a gradual decrease in forecast accuracy.
As shown in Fig. 8(a), the NSE metrics of three forecast results are at least 0.92 during the 6-168h forecast horizons.
The difference between the two is small, not more than 0.02. Among them, the CHUP-BMA method has the best NSE
metrics. However, the advantage value gradually decreases as the forecast horizon increases. The NSE metrics of the HUP-
BMA method are better than those of the selected forecast member in most forecast horizons. From Fig. 8(b), the maximum
and mean values of MAE are 1923 and 1513 m³/s for the CHUP-BMA method, 1999 and 1582 m³/s for the HUP-BMA
method, and 2179 and 1719 m³/s for the selected forecast member, respectively. The CHUP-BMA method has the best MAE
metric, with the maximum and average reduction of 10.69% and 4.36% relative to the HUP-BMA method, respectively.
Meanwhile, the MAE values of two ensemble forecasting methods are lower than those of the selected forecast members. As
shown in Fig. 8(c), the maximum and mean of the RE metric are 0.02% and -0.27% for the CHUP-BMA method, 2.97% and
1.36% for the HUP-BMA method, and 1.20% and 0.34% for the selected forecast member, respectively. The CHUP-BMA
method can reduce the RE metrics of the selected forecast member in most forecast horizons, while the HUP-BMA method
has no advantage in the RE metric.
Overall, ensemble forecast methods can somewhat improve the selected best member forecast accuracy. The CHUP-
BMA method's expectation forecast has the best accuracy, which indicates that the copula-based CHUP-BMA method can
improve the performance of the HUP-BMA method in correcting errors.
To further analyze the accuracy of ensemble forecast methods, seven floods with peaks exceeding 50,000 m³/s during
the 24 and 168h forecast horizons in the validation period (2017-2021) are selected for analyzing. The average relative error
metric of peak (PRE) (Cui et al., 2022) is added to analyze the forecasting performance for flood peaks.
Table 4 demonstrates the forecast evaluation metrics for the seven flood events. With the increase in the forecast
horizon, the NSE metric shows a decreasing trend, and the RE and MAE metrics show an increasing trend, indicating a
gradual decrease in forecasting performance. It can be seen from Table 4 that (1) in the 24h forecast horizon, the forecast
accuracy of the two methods is similar for most flood events and quality metrics, (2) in the 168h forecast horizon, the
forecast accuracy of the CHUP-BMA method is better than HUP-BMA method in most flood events and quality metrics. The
average values of the NSE, RE, MAE, and PRE are 0.88, -0.63%, 2980m³/s, and -4.55% for CHUP-BMA, and 0.84, -2.38%,
3188m³/s, and -6.46% for HUP-BMA, respectively, indicating an overall improvement of CHUP-BMA over HUP-BMA in
forecasting accuracy.
To further demonstrate the accuracy of flood process forecasting and applicability of the two methods, four relatively
large flood events are selected for comparative analysis for 168h forecast horizon (Fig. 9).
In the 20180703-flood event (Fig. 9a), the two methods have similar forecast performance, underestimating the peak
and rising water processes and overestimating the receding water process. The CHUP-BMA method has relatively low PRE
values and total runoff error. The HUP-BMA method accurately forecasts the peak present time.






**Table 4** Evaluation metrics for forecast flood events for 24 and 168h forecast horizons

| Flood event | Method | Forecast horizon (h) | Evaluation metric | | | |
|---|---|---|---|---|---|---|
| | | | NSE | RE (%) | MAE (m³/s) | PRE (%) |
| 20180703 (2018/7/1-7/7) | HUP-BMA | 24 | 0.93 | 1.95 | 1697 | -3.29 |
| | | 168 | 0.80 | 1.69 | 2709 | -8.60 |
| | CHUP-BMA | 24 | 0.94 | 3.63 | 1667 | 1.64 |
| | | 168 | 0.78 | 1.30 | 2988 | -6.26 |
| 20180714 (2018/7/11-7/17) | HUP-BMA | 24 | 0.85 | -1.38 | 2768 | -8.04 |
| | | 168 | 0.97 | 0.11 | 1101 | 0.88 |
| | CHUP-BMA | 24 | 0.84 | -1.97 | 2874 | -7.70 |
| | | 168 | 0.95 | -2.37 | 1587 | -1.23 |
| 20200717 (2020/7/14-7/20) | HUP-BMA | 24 | 0.91 | -7.02 | 3094 | -10.02 |
| | | 168 | 0.64 | -11.67 | 5965 | -19.00 |
| | CHUP-BMA | 24 | 0.91 | -4.75 | 3211 | -8.80 |
| | | 168 | 0.75 | -7.45 | 5255 | -13.58 |
| 20200727 (2020/7/25-7/31) | HUP-BMA | 24 | 0.97 | -0.22 | 1371 | 0.02 |
| | | 168 | 0.84 | -4.73 | 3044 | -13.47 |
| | CHUP-BMA | 24 | 0.94 | 4.40 | 1819 | 3.62 |
| | | 168 | 0.88 | 0.04 | 3155 | -7.79 |
| 20200815 (2020/8/12-8/17) | HUP-BMA | 24 | 0.93 | -1.31 | 2714 | -8.21 |
| | | 168 | 0.94 | -1.96 | 2259 | -9.25 |
| | CHUP-BMA | 24 | 0.96 | 2.06 | 2062 | -3.53 |
| | | 168 | 0.95 | 3.05 | 2167 | -3.82 |
| 20200820 (2020/8/18-8/24) | HUP-BMA | 24 | 0.95 | -0.79 | 2772 | 0.22 |
| | | 168 | 0.92 | 5.74 | 3509 | 11.72 |
| | CHUP-BMA | 24 | 0.96 | 2.58 | 2125 | 2.60 |
| | | 168 | 0.96 | 4.08 | 2816 | 6.06 |
| 20210907 (2021/9/4-9/10) | HUP-BMA | 24 | 0.94 | -3.26 | 2231 | -7.43 |
| | | 168 | 0.87 | -4.66 | 3042 | -13.15 |
| | CHUP-BMA | 24 | 0.97 | -0.64 | 1722 | -4.07 |
| | | 168 | 0.94 | -0.99 | 2016 | -6.82 |






In the 20200815-flood event (Fig. 9b), two methods underestimate the flood peak and overestimate the receding water
process. The HUP-BMA method has a larger flood peak error, and the CHUP-BMA method has a better fitting performance.
In the 20200820-flood event (Fig. 9c), two methods overestimate the observed flood process, with the CHUP-BMA
method having the lower peak and total runoff error than the HUP-BMA method.
In the 20210907-flood event (Fig. 9d), the CHUP-BMA and HUP-BMA methods underestimate the flood peak and
delay the forecast peak occurring time. The former has smaller peak and water volume error.

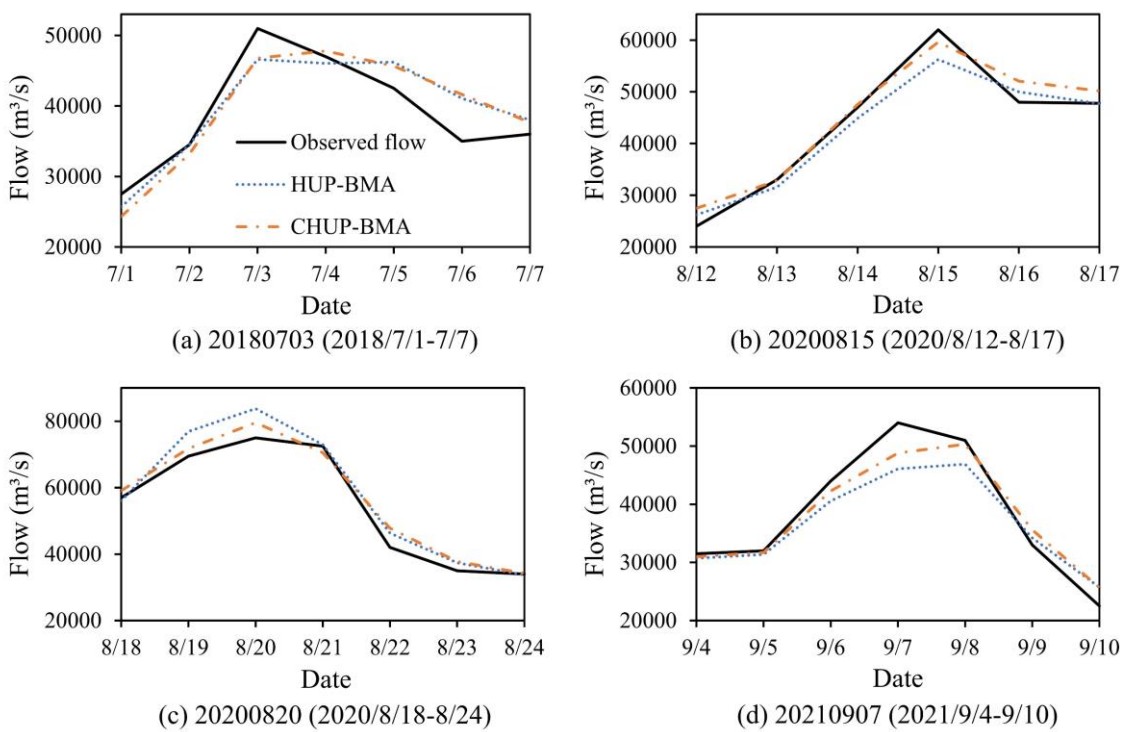

**Fig. 9** Forecasted flood events during 168h forecast horizon for the HUP-BMA and the CHUP-BMA methods

### 4.2.4 Probabilistic forecast results of ensemble forecast

**(1) Evaluation of forecast interval**

Figs. 10 (a), (b), and (c) show the CR, IW, and PUCI metrics for the forecast interval with a 90% confidence level,
respectively. Fig. 10(a) shows that during the 6-168h forecasting period, the maximum, minimum, and mean of the CR
metric for the forecast interval of the CHUP-BMA method are 0.92, 0.88, and 0.89, respectively, and 0.93, 0.88, and 0.91 for
the HUP-BMA method, respectively. The CR values of the two methods' forecast intervals are close to or exceed the 90%
confidence level, indicating that the forecast intervals are reliable.





It is obvious from Fig. 10(b) that the forecast interval width tends to increase with the increase of the forecast horizon,
indicating that the forecast uncertainty gradually increases. The maximum, minimum, and mean of the IW metrics for the
forecast interval of the CHUP-BMA method are 7820, 3337, and 6257 m³/s, respectively, and 8888, 4662, and 7345 m³/s for
the HUP-BMA method, respectively. The forecast intervals of the CHUP-BMA method are significantly narrower than those
of the HUP-BMA method, with the maximum and average reduction of 28.42% and 15.32%, respectively, which indicates
that the CHUP-BMA method can effectively reduce the interval width and forecast uncertainty.

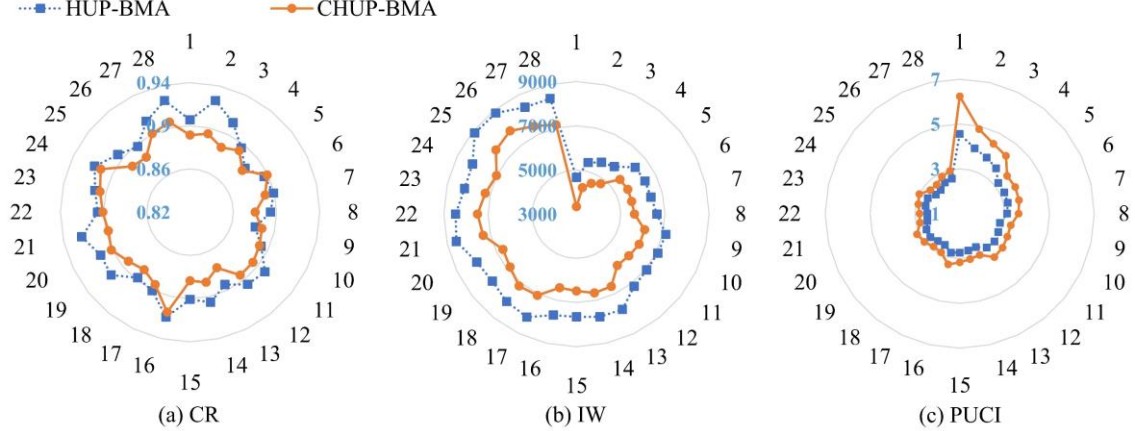

**Fig. 10** Evaluation metrics of forecast intervals with the 90% confidence level of the HUP-BMA and CHUP-BMA methods

From Fig. 10(c), the maximum, minimum, and mean of the PUCI metric for the forecast interval of the CHUP-BMA
method are 6.24, 2.65, and 3.48, respectively, and 4.55, 2.35, and 2.95 for the HUP-BMA method, respectively. The CHUP-
BMA method has the higher PUCI values, indicating that the forecast interval of the CHUP-BMA method reflects the
forecast uncertainty relatively well.
In summary, the CHUP-BMA outperforms the HUP-BMA method under the premise that the CR values are close to or
exceed the 90% confidence level. The CHUP-BMA method has narrower forecast intervals and better performance in
quantifying forecast uncertainty. Although the HUP-BMA method has a higher CR value, its IW value is larger, and the
PUCI value is smaller for the long forecast horizon, indicating that the forecast interval is too conservative to estimate the
uncertainty range reasonably.
In order to visually analyze the ability of the CHUP-BMA method to quantify forecast uncertainty, the forecast intervals
with a 90% confidence level of the HUP-BMA and CHUP-BMA methods for 168h forecast horizon in the 2020 flood season
are compared. It can be seen from Fig. 11 that the forecast intervals of the two ensemble forecasts can cover most of the
observed flows and always cover the annual maximum flood peak, indicating that the forecast intervals are reliable.
Meanwhile, the forecast intervals of the CHUP-BMA method are remarkably narrower than those of the HUP-BMA method,





indicating that the forecast uncertainty of the former is relatively low, which can provide more reasonable risk information
for TGR flood control decisions.

**Fig. 11** Forecast intervals with the 90% confidence level for the HUP-BMA and CHUP-BMA methods from 2020/7/1 8:00 to 9/24 8:00


**(2) Evaluation of overall probabilistic forecast**
Fig. 12 shows the Q-Q plots of the HUP-BMA and CHUP-BMA methods for the 24, 96, and 168h forecast horizons.
The Q-Q curves of the CHUP-BMA method are closer to the 1:1 line, indicating that the probabilistic forecasts are the more
reliable. Meanwhile, the Q-Q curves of both methods are skewed to the lower right of the 1:1 line, indicating that the
forecasts are slightly overestimated, especially for HUP-BMA method.
Meanwhile, Fig. 13 (a), (b), and (c) show the evaluation metrics of reliability ($\alpha\_index$), concentration (IGS), and
overall performance (CRPS) for the two ensemble probabilistic forecasts, respectively. It can be seen from Fig. 13(a) that the
$\alpha\_index$ metrics of the CHUP-BMA method-based probabilistic forecasts are significantly higher than those of the HUP-
BMA method in the 6-168h forecast horizons. Among them, the maximum, minimum, and mean of the $\alpha\_index$ metric for
CHUP-BMA method-based probabilistic forecasts are 0.98, 0.93, and 0.97, respectively, and 0.95, 0.88, and 0.93 for the





HUP-BMA method, respectively. The α_index metric of the CHUP-BMA method-based probabilistic forecast is closer to the
perfect value of 1, indicating that its probability forecast is the more reliable.

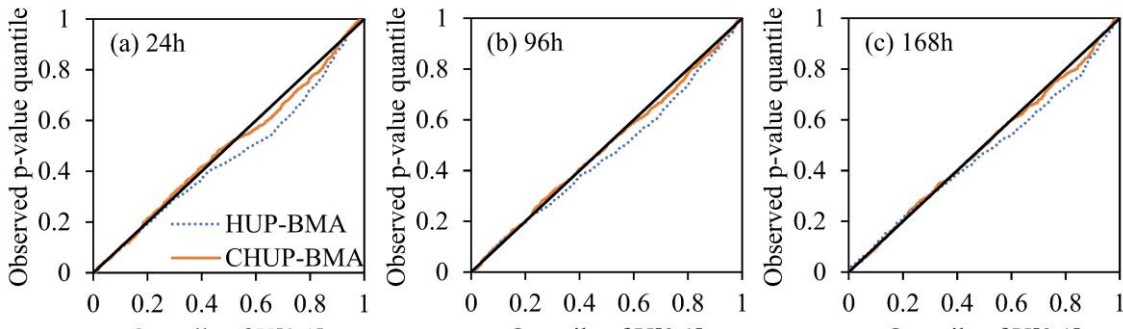


**Fig. 12** Q-Q plots of the HUP-BMA and CHUP-BMA methods for the ensemble forecasts of the 24, 96, and 168h forecast horizons. U[0,1]
denotes the uniform distribution on the interval [0,1].

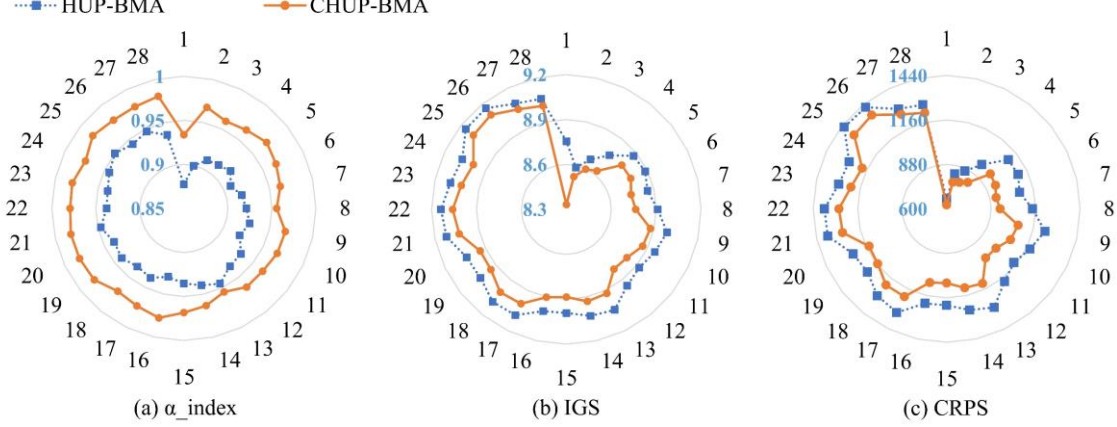


**Fig. 13** Evaluation metrics of reliability, sharpness, and overall performance of two ensemble forecasts

It can be seen from Fig. 13(b) that the IGS values of the two methods gradually increase with the increase of the
forecast horizon, indicating that the forecast uncertainty gradually increases. The maximum, minimum, and mean of the IGS
metric for the CHUP-BMA method are 9.10, 8.33, and 8.87, respectively, and 9.16, 8.59, and 8.98 for the HUP-BMA
method, respectively. It can be seen that the IGS metrics of the CHUP-BMA method are consistently lower than those of the
HUP-BMA method, which indicates that the CHUP-BMA method has a superior sharpness, assigns a high probability
density around the actual values, and has a low forecast uncertainty.
As shown in Fig. 13(c), the CRPS values of the two methods are lower than the MAE values of the selected member
(Fig. 8(b)), indicating that the probabilistic forecasts are effective and can fit the probabilistic distribution of the target values



well. Meanwhile, during the 6-168h forecast horizons, the maximum, minimum, and mean of the CRPS metric for the
CHUP-BMA method are 1356, 625, and 1074 m³/s, respectively, and 1425, 662, and 1188 m³/s for the HUP-BMA method,
respectively. It can be seen that the CRPS values of the CHUP-BMA method are lower than those of the HUP-BMA method,
with a maximum and average reduction of 17.86% and 9.71%, respectively. It can be seen that the CHUP-BMA method can
better fit the posterior distribution of the actual values and effectively improve the probabilistic forecast performance of the
HUP-BMA method.
In summary, the CHUP-BMA method considers the influence of the initial state on the ensemble forecast, bypasses the
normal quantile transformation of the HUP-BMA method, derives the posterior distribution of the target flow without
restrictions, and improves the probabilistic forecast performance of the HUP-BMA method. Therefore, the ensemble
forecasting by CHUP-BMA method can provide more reasonable and reliable risk information for the TGR.

**5 Conclusion and prospects**

This study proposed the CHUP-BMA method by coupling the copula-based HUP with the BMA method and
established an ensemble forecast scheme that consists of two forecasted precipitation, two hydrological models, and two
objective functions of parameter calibration. The ensemble forecasting performance of the HUP-BMA and CHUP-BMA
methods is discussed from the perspective of deterministic and probabilistic forecasts. The flood ensemble forecasting
experiment with 6-168h forecast horizons is conducted in the Xiangjiaba-TGR dam-site interval basin. The main conclusions
are summarized as follows.
(1) The two ensemble forecasting methods can improve the members' forecast accuracy. The proposed CHUP-BMA
method performs better than the HUP-BMA method, and the MAE metric value is reduced by a maximum of 10.69%.
(2) The coverage rate of the forecast interval of the CHUP-BMA method is close to or exceeds the specified 90%
confidence level, and the forecast interval is significantly narrower than that of the HUP-BMA method, with a maximum
reduction of 28.42%, which can effectively reduce the forecast uncertainty.
(3) The probabilistic forecast of the CHUP-BMA method has better reliability and sharpness, and its CRPS values are
reduced by a maximum of 17.86% relative to the HUP-BMA method, which indicates that the CHUP-BMA method can
better fit the posterior distribution of the actual values.
(4) The CHUP-BMA method can derive the posterior distribution of the target flow without restriction under the
condition of considering the initial constraint, which makes the BMA method more towards perfection. Therefore, it is more
suitable for the flood forecasting in the 6-168h forecast horizons and provides reliable risk information for reservoir
scheduling decision-making.
The present study focuses on flood ensemble forecasting for the TGR's 6-168h forecast horizons. Future studies can
explore the ensemble forecasting performance of the proposed CHUP-BMA method for longer forecast horizons and further
validate the effectiveness of the proposed method in global basins. Meanwhile, the effective way or method of guiding



reservoir scheduling based on ensemble forecasts can be further explored so that ensemble forecasts can be widely used in decision-making.

**Code availability**

The code used to support the findings of this study are available from the corresponding author upon request.

**Data availability**

The data generated and/or analyzed during the current study are not publicly available for legal/ethical reasons but are available from the corresponding author on reasonable request.

**Author contributions**

Zhen Cui and Shenglian Guo conceived and designed the experiments; Zhen Cui performed the experiments and wrote the manuscript draft; Zhen Cui, Shenglian Guo, Chong-Yu Xu, Hua Chen, Dedi Liu, and Yanlai Zhou reviewed and edited the manuscript.

**Competing interests**

The authors declare that they have no conflict of interest.

**Acknowledgments**

This study was financially supported by the National Key Research and Development Program of China (2022YFC3202801, 2021YFC3200305), and China Three Gorges Cooperation (0799254).

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
