# Peer review of "Quantify and reduce flood forecast uncertainty by the CHUP-BMA method"

_Hydrology and Earth System Sciences, 2023_

## Author Comment (AC1)

**Legend**

Reviewers' comments

Authors' responses

Direct quotes from the revised manuscript

**Reviewer #1:**

The paper proposes a new CHUP-BMA ensemble forecasting method by incorporating the CHUP-derived posterior distribution of the observed flow into the BMA framework. It has the advantage that the initial state constraints can be considered in the BMA while avoiding the normal quantile transformation of the HUP-BMA method. Based on deep learning, an ensemble forecasting scheme considering input, model structure, and parameter uncertainty is constructed in Three Gorges Reservoir, China, and the effectiveness of the CHUP-BMA method in reducing forecast uncertainty is verified. The study is innovative and theoretically rigorous and has promising results with solid application potential. Some questions need further discussion.

Response: We deeply appreciate your constructive comments and the time you spent on reviewing the paper. We have accepted all the revision comments. Point-by-point replies to the comments or suggestions made can be found below.

1. The sources of Figure 3 and Table 2 need to be explained to improve the reasonableness of the paper.

Response: The values of Figure 3 and Table 2 are obtained from the Hydrological Bureau of the Changjiang Water Resources Commission (HBCWRC). In addition, Figure 3 can be referred to the paper by Zhong et al. (2018b). We make the following changes in subsection 3.2.1:

The rainfall-runoff relationship graph method (Fedora and Beschta, 1989) commonly used in the Yangtze River basin can calculate the effective precipitation. The antecedent precipitation index, which is the key variable of the method, can be calculated by the

following equation to represent the soil moisture content (Zhong et al., 2018b).

$$P_{a,t+1} = k(P_{a,t} + P_t) \qquad (16)$$

$$P_{a,t+1} \le I_m \qquad (17)$$

where $P_a$ denotes the antecedent precipitation index, $P_t$ is the daily precipitation, $I_m$ is the water storage capacity of the basin, $k$ denotes evaporation reduction index.

The values of $k$ and $I_m$ for these three sub-basins are listed in Table 2, which are obtained from the Hydrological Bureau of the Changjiang Water Resources Commission (HBCWRC). Since the rainfall-runoff relationship graph method have been widely used for runoff generation calculation in the Yangtze River basin, the rainfall-runoff relationship between Xiangjiaba and Three Gorges Dam-site uncontrolled interval basin are established and plotted in Fig. 3, which is used to calculate the effective precipitation based on the antecedent precipitation index ($P_a$) and observed (or forecasted) precipitation for these three sub-basins.

**Reference**

Zhong, Y., Guo, S., Liu, Z., Wang, Y., and Yin, J. Quantifying differences between reservoir inflows and dam site floods using frequency and risk analysis methods. Stoch. Environ. Res. Risk Assess., 32, 419-433. https://doi.org/10.1007/s00477-017-1401-4, 2018b.

2. Various model inputs (e.g., rainfall, tributary flows, etc.) exist in the interval basins. The article only considers the input uncertainty of rainfall, and it is suggested to add a reason for this in subsection 3.2.1.

Response: Thanks to the reviewers for the constructive comments. There are five flow discharge inputs from five large tributaries (Jinsha, Min, Jialing, Tuo, and Wu Rivers) in our case study. The flow discharges are observed at the Pingshan, Gaochang, Fushun, Beibei, and Wulong hydrological controlled stations, respectively. Since these observed (or forecasted) flows are respectively regulated by their upstream cascade reservoirs, these flow data inputs are more accurate than the rainfall inputs.

We collected the forecasted precipitation data from the European Centre for

Medium-Range Weather Forecasts (ECMWF) and HBCWRC during the 2017-2021 flood season in the three sub-basins. Since the rainfall data is more diverse and has relatively large uncertainty, so the forecast rainfall input variable is used to explore the impact of forecast rainfall uncertainty on the Three Gorges reservoir inflow forecasts. We also make the following changes in subsection 3.2.1:

There are five flow discharge inputs from five large tributaries (Jinsha, Min, Jialing, Tuo, and Wu Rivers) in this case study. The flow discharges are observed at the Pingshan, Gaochang, Fushun, Beibei, and Wulong hydrological controlled stations, respectively. Since these observed (or forecasted) flows are respectively regulated by their upstream cascade reservoirs, these flow data inputs are more accurate than the rainfall inputs. This study collected the forecasted precipitation data from the European Centre for Medium-Range Weather Forecasts (ECMWF) and HBCWRC during the 2017-2021 flood season in these three sub-basins. Since the rainfall data is more diverse and has relatively large uncertainty, only the forecast rainfall input variable is used to explore the impact of forecast rainfall uncertainty on the Three Gorges reservoir inflow forecasts.

3. Line 255. You should briefly introduce the LSTM in subsection 3.2.2 to improve the paper's readability. In addition, it is recommended to cite references more relevant to the LSTM.

Response: Thanks for your valuable comments. We will add a brief introduction to LSTM neural networks and cite more relevant literatures as follows:

The structure of LSTM neural network includes forgetting gate, input gate, updating the state of the memory unit, and output gate (Hochreiter and Schmidhuber, 1997). The forgetting gate can select the relatively important information in the previous memory unit. The input gate can select useful information from the input variables at the current moment. The memory unit state can store relatively important information extracted from historical moments, which is updated under the control of the forgetting gate and

the input gate. The output gate selects and outputs useful information from the memory cell state. More detailed procedures of the LSTM neural network formulation have been described by Kratzert et al. (2018).

**Additional references:**

Hochreiter, S., Schmidhuber, J. Long short-term memory. *Neural Computation*, 9(8), 1735-1780. https://doi.org/10.1162/neco.1997.9.8.1735, 1997.

Kratzert, F., Klotz, D., Brenner, C., Schulz, K., Herrnegger, M. Rainfall–runoff modelling using long short-term memory (LSTM) networks. *Hydrology and Earth System Sciences*, 22(11), 6005-6022. https://doi.org/10.5194/hess-22-6005-2018, 2018.

4. Deep learning parameters significantly impact forecast accuracy, so it is recommended to show the values of deep learning parameters. The study should concentrate on ensemble forecasting methods rather than deep learning models. Therefore, the model parameter values can be shown in the appendix.

Response: Thanks for your valuable comments. We supplement the appendix with model parameter values for the ensemble members. An appendix will be added in the revised manuscript as follow:

**Appendix:**

We set the number of neural network layers and neurons to be the same for the encoding and decoding processes, with trial-and-error preferences for the number of hidden layers, neurons, and dropout. Meanwhile, the batch size, epoch, and learning rate are set to 100, 500, and 0.001, respectively. The different model parameters are shown in Table A.

**Table A** The different model parameters for ensemble membership

| Ensemble member type | Neuron | Hidden layers | Dropout |
| --- | --- | --- | --- |
| ECMWF&DA-LSTM-RED&MSE | 64 | 1 | 0.001 |
| ECMWF&LSTM-RED&MSE | 64 | 1 | 0.001 |

| | | | |
|---|---|---|---|
| ECMWF&DA-LSTM-RED&MAE | 32 | 1 | 0.01 |
| ECMWF&LSTM-RED&MAE | 64 | 1 | 0.1 |
| HBYRWRC &DA-LSTM-RED&MSE | 32 | 1 | 0.1 |
| HBYRWRC &LSTM-RED&MSE | 32 | 1 | 0.001 |
| HBYRWRC &DA-LSTM-RED&MAE | 64 | 1 | 0.001 |
| HBYRWRC &LSTM-RED&MAE | 48 | 1 | 0.01 |

5. Line 369, add a description of the member type with better forecast accuracy, i.e., the input composition, the model structure, and the objective function of the selected parameters.

Response: Thanks for your insightful comments. We have added relevant content to the article:

The member with relatively optimal forecast accuracy is composed of the forecast rainfall from ECWMF, the DA-LSTM-RED model, and the objective function with mean square error to optimize the parameters.

6. There are numerous evaluation metrics in deterministic and probabilistic forecasting. Briefly explain the reasons for the metrics chosen in the paper.

Response: Thanks for your suggestion. The article is added below:

The Nash-Sutcliffe efficiency (NSE) is one of the most important metrics in flood forecasting, reflecting the degree of fit between forecasted and observed flows (Nash & Sutcliffe, 1970). Since the accurate runoff volume predictions is more important than peak discharge for the operation of a large reservoir (Cui et al., 2023), the relative error for total runoff volume (RE) is also chosen. The mean absolute error (MAE) can reflect the forecast error for each moment, and compared with the continuous ranked probability score (CRPS) of the ensemble forecast (Raftery et al., 2005), which can reflect the effectiveness of the ensemble forecast correction. Therefore, three metrics, NSE, RE, and MAE, are selected to evaluate the deterministic forecast results.

The average coverage rate (CR) is one of the most necessary metrics for evaluating the

reliability of forecast intervals (Li et al., 2021). The average interval width (IW) is the metric that directly reflects the level of forecast uncertainty, which is an important metric for evaluating the effectiveness of the proposed methods. The percentage of observations bracketed by the unit confidence Interval (PUCI) is a comprehensive metric for evaluating the performance of forecast intervals in quantifying uncertainty (Xiong et al., 2009). Therefore, the CR, RB, and PUCI metrics are selected to evaluate the forecast intervals performance.

The $\alpha\_index$ metric can quantitatively assess the reliability of ensemble probabilistic forecasts from the perspective of distribution function values (Renard et al., 2010). The ignorance score (IGS) metric can quantitatively assess the sharpness of the posterior density function and quantify forecast uncertainty from the perspective of the probability density (Gneiting et al., 2005). The continuous ranked probability score (CRPS) is one of the most important composite metrices for assessing the overall performance of probabilistic forecasts (Raftery et al., 2005) and can represent both the reliability and sharpness of forecasted posterior distribution function. Therefore, the $\alpha\_index$, IGS, and CRPS metrics are selected to evaluate the probabilistic forecast performance.

**Additional references:**

Li, D., Marshall, L., Liang, Z., Sharma, A., Zhou, Y. Bayesian LSTM with stochastic variational inference for estimating model uncertainty in process-based hydrological models. *Water Resources Research*, 57(9), e2021WR029772. https://doi.org/10.1029/2021WR029772, 2021.

Xiong, L., Wan, M. I. N., Wei, X., O'connor, K. M. Indices for assessing the prediction bounds of hydrological models and application by generalised likelihood uncertainty estimation. *Hydrological sciences journal*, 54(5), 852-871. https://doi.org/10.1623/hysj.54.5.852, 2009.

7. Line 465, replacing 'concentration' with 'sharpness' as 'reliability (α_index), concentration (IGS),' should correspond to the name of Figure 13.

Response: Thank you for reminder. We have changed "concentration" to "sharpness" in

the revised manuscript.

8. To improve modeling rationality, explain why observations are used as model tributary inputs in training and validation periods.

Response: Thanks for your well-considered suggestions. Three Gorges Reservoir (TGR) is the largest hydraulic project in the world and controls a watershed area of 1 million $km^2$. There are more than 300 larger-scale reservoirs have been built in the upstream Yangtze River basin with a total storage of 163.3 billion $m^3$. The operational flow forecasting procedure is from the sources of tributaries, each larger-scale reservoir inflows, interval basin flow forecasting and river flow routing to downstream sections, and so on. Since the forecast data series at the outlets of five large tributaries (Jinsha, Min, Jialing, Tuo, and Wu Rivers) are inconsistent, we used the observed flows to train and validate the proposed models or forecasting schemes. We have added the following in subsection 3.2.2:

In the actual TGR inflow forecasting model, the observed flow discharge data at the outlets of five large tributaries (Jinsha, Min, Jialing, Tuo, and Wu Rivers) in the interval-basin between Xiangjiaba and TGR dam-site are used to train and validate the proposed models or forecasting schemes since the forecast data series at the outlets of tributaries are inconsistent.

9. In the outlook, adding the construction of the CHUP-BMA method using a more flexible vine copula will make the CHUP-BMA method more competitive.

Response: Thank you for your foresighted suggestions. Our additions to the article are as follows:

In the future, the vine copula, which facilitates multivariate joint distribution modelling, can be considered for constructing the CHUP-BMA method and exploring its advantages and effectiveness in ensemble flood forecasting.

---

## Author Comment (AC2)

**Reply to CC1' comments**

**Legend**

Reviewers' comments

Authors' responses

Direct quotes from the revised manuscript

Cui et al. (2023) mainly proposed a CHUP-BMA method to solve the unreasonable assumption of normal distribution of the BMA framework in hydrological forecast field. This specialized theory has been applied to the Three Gorges region of China to demonstrate its feasibility. The study is interesting and meaningful to the hydrological forecast community. However, it needs some revision before it is up to the publication standard of HESS.

Response: We deeply appreciate your constructive comments and the time you spent on reviewing the paper. We have accepted all the revision comments. Point-by-point replies to the comments or suggestions made can be found below.

1. Lines 8-10, the statement is not accurate. As I know, few existing literature (with Copula tool or without it) has been devoted to avoiding the normal transformation in the HUP-BMA method.

Response: After further careful review by the authors, it is found that this paper is indeed the first article used to solve the problem of the normal distribution assumption of the HUP-BMA method. In order to be more rigorous, the corresponding content is modified to "This study introduced a copula-based HUP in the framework of BMA and proposed the CHUP-BMA method to bypass the need for normal quantile transformation of the HUP-BMA method".

2. Lines 68-70 are not clear. It is ambiguous that " When the member forecasts are the same, the ensemble forecasts produce the same conditional probability distribution and lack rationality". The parameters of the BMA method include membership weights and variances, and the posterior distribution of the ensemble forecast is not necessarily the same even if the forecast members have the same results. In order to reflect the necessity of the initial state, the article should be changed to " When the forecast results of a member are the same at different moments, the same forecast conditional probability distribution will be generated, which is not reasonable." It is important to highlight that the distribution is the same at different moments.

Response: Thank you very much for your constructive comments. It has been changed to " When the forecast results of a member are the same at different moments, the same forecast conditional probability distribution will be generated, which is not reasonable." in the paper.

3. Line 79, missing punctuation.

Response: The punctuation has been added to the article.

4. Line 120, unit superscript error.

Response: Error subscripts have been modified.

5. Line 186, the symbol $c_m()$ does not appear in Eq. (12).

Response: The $c_2$ in Eq. (12) denotes the 2-dimensional copula density function. $c_m$ denotes the m-dimensional copula density function. To reduce ambiguity, it has been supplemented with "$m$ denotes the dimension."

6. Line 279 is not clear. To improve the readability and logic of the paper, it is suggested to revise as "the forecasted flow of the upstream mainstream station".

Response: The corresponding content has been revised to "the forecasted flow of the upstream mainstream station".

7. Line 339, whether these distributions and Copula functions passed the K-S test or other assumption tests?

Response:

[Figure]

Fig.1 The KS statistics of $Q_o$, $Q_b$, and $Q_f$ sequence marginal distributions and copula functions. 1, 2, … ,28 denote 6h, 12h, …, 168h forecast horizons, respectively

Fig.1 illustrates the average of the KS statistics for the eight members. The Lognormal, Gamma and P-III passed the K-S test for marginal distribution. The Student, Gaussian, and Frank copula passed the K-S test for copula function. The Lognormal and Student copula have relatively low KS statistics, which follows the same trend as the RMSE values. Therefore, it is found that both the K-S test and the RMSE criterion are effective in reflecting the fitting performance of the marginal distribution and copula function. To save space in the article, only the RMSE values are shown.

8. Line 498, although the authors do not mention it, it is necessary to mention the improvement room of the inherent mechanism of the CHUP-BMA method.

Response: Thanks for the valuable comments. The improvement room of CHUP-BMA method has been added in the article "In this study, the copula-based HUP is coupled with the BMA method, and the CHUP-BMA method is proposed, which not only can consider the influence of the initial state on the ensemble forecast, but also can avoid the assumption of normal distribution in the HUP-

BMA method and derive the posterior distribution function more accurately. An ensemble forecast scheme that consists of two forecasted precipitation, two hydrological models, and two objective functions of parameter calibration is established".

---

## Author Comment (AC3)

**Reply to CC2' comments**

**Legend**

Reviewers' comments

Authors' responses

Direct quotes from the revised manuscript

The paper couples the Copula-based hydrological uncertainty processor with the Bayesian model averaging method to quantify and reduce uncertainty in flood forecasting upstream of the Three Gorges Reservoir in the Yangtze River basin, China. The topic is timely and the paper is technically sound. The paper could benefit from additional clarification in some sections.

Response: We deeply appreciate your constructive comments and the time you spent on reviewing the paper. We have accepted all the revision comments. Point-by-point replies to the comments or suggestions made can be found below.

1. Line 9: The full name of "CHUP-BMA" needs to be given the first time it is mentioned.

**Response:** The corresponding content has modified to "This study proposed the CHUP-BMA method by introducing a copula-based HUP in the framework of BMA to bypass the need for normal quantile transformation of the HUP-BMA method".

2. Lines 68-70: The description here is not clear to me, e.g. why the BMA ignores the constraint of initial conditions. Further explanation of the reason and how the HUP-BMA mentioned later can obtain the posterior distribution function of the observed flow is suggested in the Introduction section.

**Response:** It can be shown from Raftery et al. (2005) that the conditional distribution of the member ($Q_{f,i}$) in the BMA is assumed to follow the normal distribution with expectation $\mu_i = a_i + b_i \cdot Q_{f,i}$ ($a_i$ and $b_i$ are the bias correction coefficients) and variance $\sigma_i$, which implies that the conditional distribution is only related to the member's forecasted flow and is not affected by the observed flow at the start of the forecast. Therefore, it is not reasonable to produce the same posterior distribution when the forecast results are the same at different moments. The corresponding content has been modified as follows:

However, most studies ignore an essential issue: the BMA does not consider the constraint of initial conditions (i.e., observed flow at the start of the forecast). It can be shown from Raftery et al. (2005) that the conditional distribution of the member ($Q_{f,i}$) in the BMA is assumed to follow the normal distribution with expectation $\mu_i = a_i + b_i \cdot Q_{f,i}$ ($a_i$ and $b_i$ are the bias correction coefficients) and variance $\sigma_i$, which implies that the conditional distribution is only related to the member's forecasted flow and is not affected by the observed flow at the start of the forecast. It is not reasonable to produce the same posterior distribution when the forecast results are the same at different moments. The hydrological uncertainty processor (HUP) can obtain the posterior distribution function of the actual value under the condition of the forecast value and the observed flow at the start of the forecast based on Bayesian principles and the assumption of perfect rainfall forecasting (Krzysztofowicz and Kelly, 2000). Darbandsari and Coulibaly (2021) firstly constructed the conditional distribution of the observed flow under the conditions of the member forecasted flow and the observed flow at the start of the forecast and used the BMA method to weight the conditional distribution of all members to obtain the final posterior distribution, which is called the HUP-BMA method. Their results showed that the HUP-BMA method outperforms the HUP method and improves the BMA method in short-term probabilistic forecasting. In addition, the derivability of the posterior distribution for the ensemble members is theoretically enhanced, the heteroskedasticity of the ensemble members is considered, and the interpretability and logical rationality of the BMA method are improved.

3. Lines 85-86: It seems that this work is motivated by the copula-based HUP method in Liu et al. I suggest giving a brief description of this method and how it is used to improve forecast accuracy here.

**Response:** A modification has been made to the article as follows:

Liu et al. (2016) adopted the copula to derive the conditional distribution of the observed flow under the conditions of the forecasted flow, which avoids the assumption that the flow series obeys a normal distribution in the HUP and relaxes the application limitation. The study shows that the CHUP can improve the probabilistic forecasting performance of the HUP method.

4. Line 87: I suggest presenting the objectives and research steps one by one. For example, the novelty of this work can be introduced in the previous paragraph, along with the shortcomings of

current methods, and then the implementation of the proposed method in streamflow forecasting can be briefly introduced.

**Response:** The corresponding content has been modified as follows:

The main innovations and research steps are shown as follows: (1) A novel CHUP-BMA method is proposed for the first time by coupling CHUP into BMA, which not only solves the problem of the flow series obeying the assumption of normal distribution in HUP-BMA, but also considers the constraints of the initial condition of the forecast. (2) An ensemble forecast containing eight members is constructed by combining two types of forecast precipitation, two long short-term memory (LSTM) models, i.e., the recursive encoder-decoder structure-based LSTM-RED model and the feature-temporal dual attention-based DA-LSTM-RED model, and two objective functions of model calibration. (3) The ensemble forecast performance of the proposed method is analyzed and discussed in comparison to the HUP-BMA benchmark method in terms of the deterministic and probabilistic forecast. The interval basin between Xiangjiaba Dam and the Three Gorges Dam is selected as case study

5. Section 3.2: It seems that the model structure uncertainty in this study is considered by using two forecast models with LSTM-RED structure. why not using two different types of models (e.g., ANN-based vs. tree-based or physical-based vs. data-driven models)?

**Response:** It has been demonstrated in many studies that LSTM models have relatively better forecasting performance than ANN, tree-based models, and physical mechanism models (Kratzert et al., 2018; Hu et al., 2018; Han & Morrison, 2021; Zhang et al., 2022; Hayder et al., 2023). Meanwhile, in the interval basin between Xiangjiaba and TGR dam-site, the physical mechanism model usually is very complex and has low forecasting accuracy. This study is a further research based on Cui et al. (2023), which demonstrated that the DA-LSTM-RED model structure improved by the dual-attention mechanism resulted in a substantial improvement in forecast accuracy relative to the LSTM-RED model. Therefore, this study uses two advanced LSTM-RED and DA-LSTM-RED models for flood forecasting and focuses on the uncertainties associated with these two models.

**Reference:**

Kratzert, F., Klotz, D., Brenner, C., Schulz, K., and Herrnegger, M.: Rainfall–runoff modelling using

Long Short-Term Memory (LSTM) networks, Hydrol. Earth Syst. Sci., 22, 6005–6022, https://doi.org/10.5194/hess-22-6005-2018, 2018.

Hu, C., Wu, Q., Li, H., Jian, S., Li, N., & Lou, Z. Deep learning with a long short-term memory networks approach for rainfall-runoff simulation[J]. Water, 10(11): 1543, https://doi.org/10.3390/w10111543, 2018.

Han, H., & Morrison, R. R. Data-driven approaches for runoff prediction using distributed data[J]. Stoch. Environ. Res. Risk. Assess., 1-19. https://doi.org/10.1007/s00477-021-01993-3, 2021.

Zhang, Y., Ragettli, S., Molnar, P., Fink, O., & Peleg, N. Generalization of an Encoder-Decoder LSTM model for flood prediction in ungauged catchments[J]. J. Hydrol., 614: 128577. https://doi.org/10.1016/j.jhydrol.2022.128577,2022.

Hayder, I. M., Al-Amiedy, T. A., Ghaban, W., Saeed, F., Nasser, M., Al-Ali, G. A., & Younis, H. A. An Intelligent Early Flood Forecasting and Prediction Leveraging Machine and Deep Learning Algorithms with Advanced Alert System. Processes, 11(2), 481. https://doi.org/10.3390/pr11020481, 2023.

Cui, Z., Guo, S., Zhou, Y., Wang, J. Exploration of dual-attention mechanism-based deep learning for multi-step-ahead flood probabilistic forecasting. J. Hydrol., 622, 129688. https://doi.org/10.1016/j.jhydrol.2023.129688, 2023.

6. Also, what is the purpose of using MAE and MSE as prediction evaluation metrics in this work? These two metrics are similar to each other. In order to account for model parameter uncertainty, it seems more appropriate to use three apparently different evaluation metrics, such as Nash-Sutcliffe efficiency, mean absolute error (MAE), and relative error of total discharge (RE).

**Response:** In this paper, we have trained the model parameters using different loss functions. The indicator MAE focus on different points, while the MAE focus on the magnitude of the error mean, and the MSE is sensitive to outliers with large errors. As a result, the model can be guided to produce varying parameters to account for parameter uncertainty. We make the following changes in the article:

For example, the mean absolute error function focuses on the magnitude of the error mean. The mean square error function usually is sensitive to outliers with large errors, which may make the model parameters with different objective functions produce forecast results with different focus

points (Duan et al., 2007).

7. Some reference to the first mentioned methods is suggested, e.g. the reference to the "Adam method" is suggested in line 283.

**Response:** We make the following changes:

The model is trained by the Adam method (Kingma& Ba, 2014).

**Additional references:**

Kingma D P, Ba J. Adam: A method for stochastic optimization[J]. arXiv preprint arXiv:1412.6980, https://doi.org/10.48550/arXiv.1412.6980, 2014.

---

## Author Comment (AC4)

**Reply to Reviewers' comments (Reviewer#2)**

**Legend**

Reviewers' comments

Authors' responses

Direct quotes from the revised manuscript

**Reviewer #2:**

The authors combine the copula-based hydrological uncertainty processor (CHUP) and Bayesian model averaging (BMA) to obtain a novel approach to statistical post-processing of hydrological ensemble forecasts. The proposed approach is promising and the presented results are fair, but the paper needs some improvement and it also raises some questions.

Response: We deeply appreciate your constructive comments and the time you spent on reviewing the paper. We have accepted all the revision comments. Point-by-point replies to the comments or suggestions made can be found below.

Major comments:

1. L28: The cited paper Sloughter et al. (2010) deals with post-processing wind speed forecasts. The BMA model for precipitation is introduced in Sloughter et al. (2007).

Response: Firstly, thank you very much for your careful and detailed suggestions. We have found the equivalent in line 48 and have made the following supplementary revisions:

The BMA method is initially successfully applied to the ensemble forecast of meteorological elements such as temperature, precipitation, and wind speed (Raftery et al 2005; Sloughter et al, 2007; Sloughter et al, 2010).

**Additional references:**

Sloughter, J. M., Raftery, A. E., Gneiting, T. and Fraley, C. (2007) Probabilistic quantitative precipitation forecasting using Bayesian model averaging. Mon. Weather Rev. 135, 3209–3220.

2. I am also missing references to BMA models for hydrological forecasts, e.g. Hemri et al. (2013) or Baran et al. (2019).

Response: Thanks for your constructive comments. References have been added to the paper.

Hemri et al. (2013) introduced the principle of Geostatistical output perturbation (GOP) into the BMA method, and extended the membership probability distribution into a multivariate normal distribution function, proposing a multivariate BMA. Relative to the univariate BMA method, this method can not only consider the temporal correlation between forecast flows, but also improve the forecast reliability when the forecast system was changing, i.e., fewer models were available due to dropping out at particular lead times. In order to ensure that the quantiles of forecast distributions after Box-Cox transformation are within the actual physical range, Baran et al. (2013) introduced upper and lower truncated normal distributions into the BMA, they found that the double truncated BMA had reliable forecasting ability compared to ensemble model output statistics, and the advantage was more obvious when rolling window training periods are used.

**Additional references:**

Baran, S., Hemri, S. and El Ayari, M. (2019) Statistical post-processing of water level forecasts using Bayesian model averaging with doubly-truncated normal components. Water Resour. Res. 55, 3997–4013.
Hemri, S., Fundel, M. and Zappa, M. (2013) Simultaneous calibration of ensemble river flow predictions over an entire range of lead times. Water Resour. Res. 49, 6744–6755.

3. Eq.4: In the original description of the HUP, different CDFs are considered for the forecasts and the observations, moreover, in the former case it is considered as an initial estimate. Does such a relaxation make sense here as well?

Response: Thanks for your insightful suggestions. We will add the following changes to the paper.

The HUP method is a meta-Gaussian model assuming that flow series transformed to

normal space obey the Gaussian distribution. The cumulative distribution function is different for forecasted and observed flows. The common normal quantile transformation is key to the application of the HUP method, and its significance is to make the HUP method applicable to variables with any marginal distributions, heteroskedasticity, and nonlinear dependence structures (Krzysztofowicz and Kelly, 2000; Darbandsari and Coulibaly, 2021).

4. Section 3.1.2. introducing the HUP follows the structure of Sections 2.1.2 – 2.1.4 of Darbandsari and Coulibaly (2021); however, one should mention that the Markov process of Eq.5 is stationary and define exactly how θt in L169 is related to Eq.7 (see Darbandsari and Coulibaly, 2021, Eq.10).

Response: Thanks for your perceptive suggestions. We will add the following changes:

The HUP method assumes that the observed flow obeys the strictly stationary first-order Markov process (Krzysztofowicz and Kelly, 2000)

$\hat{Q}_b$, $\hat{Q}_o$, and $\hat{Q}_{f,i}$ are assumed to obey a linear relationship. The expression of the likelihood function in normal space is as follows.

$$\hat{Q}_{f,i,t} = a_t \times \hat{Q}_{o,t} + d_t \times \hat{Q}_b + b_t + \theta_t$$

$$p(\hat{Q}_{f,i,t}|\hat{Q}_{o,t}, \hat{Q}_b) = \frac{1}{\sigma_t} n \left\{ \frac{\hat{Q}_{f,i,t} - (a_t \times \hat{Q}_{o,t} + d_t \times \hat{Q}_b + b_t)}{\sigma_t} \right\} \qquad (7)$$

where, $\theta_t$ is an independent variable obeying N(0,$\sigma_t^2$). $a_t$, $d_t$, and $b_t$ are regression coefficients.

5. L310: "The IGS metric indicates the sharpness of the probabilistic forecast". The IGS, similar to the CRPS addresses simultaneously both calibration and sharpness, as indicated in the cited work of Gneiting et al. (2005). Hence, I think referring to IGS as a measure of concentration is slightly misleading.

Response: Thanks for your valuable comments. We will correct the misleading content

and make a corresponding change in line 310:

The IGS and CRPS metrics can reflect the reliability and sharpness of the probabilistic forecast. The former can quantify the forecast probability density at the observation, while the latter can indicate the fit performance between the posterior probabilistic distribution and the actual probabilistic distribution of Qo (Raftery et al., 2005). Both CRPS and IGS are negative scores, i.e., the smaller the value, the better. The IGS imposes severe penalties for particularly poor probabilistic predictions and may be extremely sensitive to outliers and extreme events, yet also lacks robustness (Raftery et al., 2005).

A corresponding change in line 465:

Meanwhile, Fig. 13 (a), (b), and (c) show the evaluation metrics of the ensemble probabilistic forecast.

A corresponding change in line 480:

It can be seen from Fig. 13(b) that the IGS values of the two methods gradually increase with the increase of the forecast horizon, indicating that the forecast uncertainty gradually increases. The maximum, minimum, and mean of the IGS metric for the CHUP-BMA method are 9.10, 8.33, and 8.87, respectively, and 9.16, 8.59, and 8.98 for the HUP-BMA method, respectively. It can be seen that the IGS metrics of the CHUP-BMA method are consistently lower than those of the HUP-BMA method, which indicates that the CHUP-BMA method has better ensemble forecast performance relative to the HUP-BMA method by assigning a higher probability density around the actual values.

6. In Section 4, I would definitely consider the corresponding scores (or at least some of them) for the ensemble forecasts as well.

Response: Thanks very much for your insightful suggestions. Some of the evaluation metrics with a high degree of acceptance corresponding to ensemble forecasts, such as the IGS and CRPS metrics, have been used in the paper, supplemented by the

probability integral transform (PIT) histogram, which is more intuitive relative to the Q-Q diagram.

[Figure]

**Fig. 12** The probability integral transform (PIT) histograms of the HUP-BMA and CHUP-BMA methods for the ensemble forecasts of the 24, 96, and 168h forecast horizons.

Fig. 12 shows the PIT histograms of the HUP-BMA and CHUP-BMA methods for 24, 96, and 168h forecast horizons. It can be significantly observed that the PIT plots of the HUP-BMA method show a ∩-shaped distribution, which indicates that the forecast distribution is over-dispersed and overestimates the forecast uncertainty, explaining the phenomenon of wide intervals. Meanwhile, the PIT plot of CHUP-BMA is more uniformly distributed than that of the HUP-BMA method, which can obtain a better calibration performance.

7. What can be said about the statistical significance of the score differences between HUP-BMA and CHUP-BMA?

Response: Thanks for your valuable suggestions. We supplemented the statistical significance of the score differences between HUP-BMA and CHUP-BMA.

Table 5 T-test results of ensemble forecast metrics at 0.05 significance level

| Metric | α_index | | IGS | | CRPS | |
|---|---|---|---|---|---|---|
| | HUP-BMA | CHUP-BMA | HUP-BMA | CHUP-BMA | HUP-BMA | CHUP-BMA |
| Mean | 0.93 | 0.97 | 8.98 | 8.87 | 1188 | 1074 |
| Variance | 0.0003 | 0.0001 | 0.02 | 0.03 | 32247 | 33716 |
| Degree of freedom | 46.00 | | 52.00 | | 54.00 | |
| T-statistic | -10.76 | | 2.36 | | 2.34 | |
| T-threshold | 1.68 | | 1.67 | | 1.67 | |
| Difference significance analysis | Significant | | Significant | | Significant | |

From the Table 5, it can be seen that the T-statistics at the 0.05 significance level for all three metrics are higher than the threshold value, indicating that there is a significant difference between the scores of the CHUP-BMA and HUP-BMA methods, i.e., the CHUP-BMA method is significantly better than the HUP-BMA method for ensemble forecasting metrics and performance.

Minor remarks, typos:

1. L205-206: "It has been studied that the BMA method with sliding windows can obtain better probabilistic forecast performance". Better compared to what?

Response: Thanks for your thoughtful suggestions for changes. The following changes have been made:

Parrish et al. (2012) and Darbandsari and Coulibaly (2019) have shown that the BMA method with the sliding window can obtain better probabilistic forecast performance compared to the method without the sliding window.

2. L307: "indicative function" → "indicator function"

Response: Thanks for your detailed suggestions for changes. The following changes have been made:

$I(\cdot)$ denotes the indicator function.

---

## Editor Decision (ED1)

**Final Comments on HESS-2023-106:**

**Quantify and reduce flood uncertainty by the CHUP-BMA method.**

Thank you very much to the authors for providing this revised version including all referee comments. I still have some minor comments on the manuscript. After these comments are addressed the paper should be ready for the final step.

Minor corrections

- Lines 15-16 in Abstract: Can you please check the wording in these two-lines? It is not clear what is the message here.

Lines 47-48: Can you please revise wording here?

-Line 239: Acronym of second model for precipitation forecast (HBCWRC) does not coincide with the acronym in Figure 2.

- Line 243-244: Please review wording. It looks like something is missing.

- Line 248: Please define $P_{a,t}$

- Figure 3: What are the units of Pa?

- Line 287: Replace "to neural network" by "to the neural network"

- Line 303: Replace" is the input variables" by "is the input variable"

- Line 311: Replace "types data" by "data types"

- Line 343: Replace "cate" by "rate"

- Line 420: Replace "expectation" by "expected"

- Figure 13: Can you include the values of $\alpha_{index}$, IGS and CRPS in parenthesis after its corresponding meaning, in the legend of the Figure 13? That will make the figure easier to read without having to back to the meaning of the metrics.

- Line 558: Can you please check the wording in this sentence?

---

## Author Response (AR2)

**Author's response**

**Dear Professor Lelys Bravo de Guenni**

We have carefully revised the manuscript according to your valuable comments and suggestions. On behalf of all authors, I am pleased to submit the revised version of our manuscript titled "Quantify and reduce flood forecast uncertainty by the CHUP-BMA method".

The manuscript has been revised along with the review suggestions. All comments have been modified or addressed in the revised version. All newly added parts (except minor language corrections) are marked in BLUE for easy review. We sincerely hope that you will find the revised version to your satisfaction. All authors have reviewed the revision and agree to the submission.

Thank you very much for your time and efforts on our manuscript again.

Best Regards,

Corresponding author
Prof. Shenglian Guo
State Key Laboratory of Water Resources Engineering and Management,
Wuhan University,
Wuhan 430072, P. R China
**E-mail:** slguo@whu.edu.cn
May 6, 2024

**Reply to Lelys Bravo de Guenni (editor)**

**Legend**

Reviewers' comments

Authors' responses

Direct quotes from the revised manuscript

**Editor:**

Thank you very much to the authors for providing this revised version including all referee comments. I still have some minor comments on the manuscript. After these comments are addressed the paper should be ready for the final step.

Response: We deeply appreciate your constructive comments and the time you spent on reviewing the paper. We have accepted all the revision comments. Point-by-point replies to the comments or suggestions made can be found below.

1. Lines 15-16 in Abstract: Can you please check the wording in these two-lines? It is not clear what is the message here.

Response: We have rewritten the sentences as following:

Compared with the HUP-BMA method, the forecast interval width and continuous ranked probability score metrics of the CHUP-BMA method are reduced by a maximum of 28.42% and 17.86% within all forecast horizons, respectively.

2. Lines 47-48: Can you please revise wording here?

Response: We have rewritten the sentences as following:

The BMA method has been applied to temperature, precipitation, and wind speed ensemble forecasts of meteorological forcing.

3. Line 239: Acronym of second model for precipitation forecast (HBCWRC) does not coincide with the acronym in Figure 2.

Response: We have replaced HBCWRC with HBYRWRC.

4. Line 243-244: Please review wording. It looks like something is missing.

Response: We have carefully checked the wording and have made the following changes.

The observed and forecasted precipitations are converted into the effective

precipitation in the three sub-basin areas, which accounts for the losses of plant reception, infiltration, evaporation, etc.

5. Line 248: Please define Pa,t

Response: We have added relevant content to the article:

$P_{a,t}$ denotes the antecedent precipitation index on the $t$-th day.

6. Figure 3: What are the units of Pa?

Response: Units of Pa have been added to Figure 3.

[Figure]

**Fig. 3** Rainfall-runoff relationship between Xiangjiaba and Three Gorges dam-site uncontrolled interval basin

7. Line 287: Replace "to neural network" by "to the neural network".

Response: We have replaced "to neural network" by "to the neural network".

8. Line 303: Replace "is the input variables" by "is the input variable".

Response: We have replaced "is the input variables" by "is the input variable".

9. Line 311: Replace "types data" by "data types".

Response: We have replaced "types data" by "data types".

10. Line 343: Replace "cate" by "rate"

Response: We have replaced "cate" by "rate".

11. Line 420: Replace "expectation" by "expected"

Response: We have replaced "expectation" by "expected".

12. Figure 13: Can you include the values of $\alpha_{index}$, IGS and CRPS in parenthesis after its corresponding meaning, in the legend of the Figure 13? That will make the figure easier to read without having to back to the meaning of the metrics.

Response: Due to the size constraints of the figure, we have added the meanings of the metrics to the figure name to make it easier to read.

[Figure]

**Fig. 13** Evaluation metrics of α_index, IGS, and CRPS metrics of two ensemble forecasts. The α_index metric can assess the reliability of ensemble forecasts, while the IGS and CRPS metrics can reflect the reliability and sharpness of the ensemble forecasts. The closer the α_index metric is to 1, and the smaller the IGS and CRPS metrics are, the better the performance of the ensemble forecast.

13. Line 558: Can you please check the wording in this sentence?

Response: We have rewritten the sentences as following:

In this study, we proposed a novel CHUP-BMA method, which not only can consider the influence of the initial state on the ensemble forecast, but also can avoid the assumption of normal distribution in the HUP-BMA method and derive the posterior distribution function more accurately.